# Intercomparison of freshwater fluxes over ocean and investigations into water budget closure

Marloes Gutenstein[1], Karsten Fennig[1], Marc Schröder[1], Tim Trent[2], Stephan Bakan[3], J. Brent Roberts[4], and Franklin R. Robertson[4]

[1]Deutscher Wetterdienst, Offenbach, Germany
[2]University of Leicester, Leicester, United Kingdom
[3]Max Planck Institute for Meteorology, Hamburg, Germany
[4]NASA Marshall Space Flight Center, Huntsville, AL, USA

**Correspondence:** Marloes Gutenstein (marloes.gutenstein@dwd.de)

**Abstract.** The development of algorithms for the retrieval of water cycle components from satellite data, such as total column water vapor content (TCWV), precipitation (P), latent heat flux, and evaporation (E) has seen much progress in the past three decades. In the present study, we compare six recent satellite-based retrieval algorithms and ERA5 (the European Centre for Medium-Range Weather Forecasts' fifth reanalysis) freshwater flux (E-P) data regarding global and regional, seasonal and inter-annual variation to assess the degree of correspondence among them. The compared data sets are recent, freely available and documented climate data records (CDRs), developed with a focus on stability and homogeneity of the time series, as opposed to instantaneous accuracy.

One main finding of our study is the agreement of global ocean means of all E-P data sets within the uncertainty ranges of satellite-based data. Regionally, however, significant differences are found among the satellite data and with ERA5. Regression analyses of regional monthly means of E, P, and E-P against the statistical median of the satellite data ensemble (SEM) show that, despite substantial differences in global E patterns, deviations among E-P data are dominated by differences in P throughout the globe. E-P differences among data sets are spatially inhomogeneous.

We observe that for ERA5 long-term global E-P is very close to 0 mm/day and that there is good agreement between land and ocean mean E-P, vertically integrated moisture flux divergence (VIMD), and global TCWV tendency. The fact that E and P are balanced globally provides an opportunity to investigate the consistency between E and P data sets. Over ocean, P (nearly) balances with E if the net transport of water vapor from ocean to land (approximated by over-ocean VIMD, i.e., $\nabla \cdot (vq)_{ocean}$) is taken into account. On a monthly time scale, linear regression of $E_{ocean} - \nabla \cdot (vq)_{ocean}$ with $P_{ocean}$ yields $R^2 = 0.86$ for ERA5, but smaller $R^2$ are found for satellite data sets.

Climatological global yearly totals of water cycle components (E, P, E-P, and net transport from ocean to land and *vice versa*) calculated from the data sets used in this study are in agreement with previous studies, with ERA5 E and P occupying the upper part of the range. Over ocean, both the spread among satellite-based E and the difference between two satellite-based P data sets are greater than E-P and these remain the largest sources of uncertainty within the observed global water budget.

We conclude that for a better understanding of the global water budget, the quality of E and P data sets needs to be improved and the uncertainties more rigorously quantified.

# 1  Introduction

The water and energy cycles are key components of Earth's climate system. Energy exchange from water phase changes plays a direct role in atmospheric heating; therefore, precipitation (P) and evaporation (E) are two critical processes connecting the land/ocean surface and overlying atmosphere (Trenberth et al., 2009). The difference between E and P rates, $E - P$, is the freshwater flux from the surface to the atmosphere, which is positive where E dominates and negative where P dominates. Over the global oceans, total $E - P$ is positive, as a considerable amount of water evaporates from the oceans and is transported to land by advection, mainly in the form of water vapor, where it precipitates. Averaged over a year, changes in atmospheric storage vanish and net negative $E - P$ over land is balanced by continental runoff of water into the ocean. Although numerous studies have addressed the question of how variations in the ocean state affect the water cycle and freshwater fluxes with a particular view on global warming (Wentz et al., 2007; Trenberth et al., 2007; Schlosser and Houser , 2007; Robertson et al., 2014), a clear and consistent picture has yet to emerge — one of the significant challenges in climate science (Bony et al., 2015; Hegerl et al., 2014; Allan et al., 2020).

At long temporal and/or large spatial scales, the increases in E and P with rising global temperature are relatively small ($2 - 3\%$ K$^{-1}$) and are constrained by the energy budget. At smaller scales (less than approximately 4000 km and/or 10 years) these changes can be much larger (or smaller) due to dynamical contributions (Dagan et al., 2019; Yin and Porporato, 2019; Allan et al., 2020). The nature and extent of these changes, which affect the livelihoods of many millions of people, are difficult to model due to various counteracting influences such as forcing by clouds and aerosols, or land use change (Allan et al., 2020). Close monitoring of E and P by (satellite) observations thus yields an important contribution to a better understanding of impacts of climate change at regional and local scales.

The study of the global water cycle is not only compelling from a scientific point of view: it also aids the evaluation of climate models and reanalyses by verifying the degree of consistency among the various components of the cycle. Such an approach is adopted here for the evaluation of satellite observations of $E$ and $P$, which, particularly over ocean, are difficult to validate otherwise. The fact that the global water cycle is closed puts a strong constraint on global total E and P fluxes. This has been exploited in various studies in the past (Trenberth et al., 2007; Schlosser and Houser , 2007; Berrisford et al. , 2011; Trenberth et al., 2011; Trenberth and Asrar , 2014; Trenberth and Fasullo, 2013; Seager and Henderson , 2013; Robertson et al., 2014) from which the general conclusion emerged that, although much progress has been made regarding $E$ and $P$ estimates, observations and models still require substantial improvements in accuracy to achieve budget closure.

Over the years, methods to determine $E$ and $P$ based (mainly) on satellite data have been developed and repeatedly updated: HOAPS E and P (Andersson et al., 2017), J-OFURO E (Tomita et al., 2019), IFREMER E (Bentamy et al., 2013), SEAFLUX E (Roberts et al., 2020), OAFlux E (Yu et al., 2008), and GPCP P (GPCP, 2018) are among the most widely used data sets. Acronyms are explained in Section 2 and listed in Table 1. We present an intercomparison of these data sets, all freely available Climate Data Records (CDRs), characterized by the stability of input data and retrieval algorithms, emphasizing data homogeneity over local, instantaneous accuracy. European Centre for Medium-range Weather Forecast (ECMWF) ERA5 reanalysis data (Hersbach et al., 2020) are included for comparison in the present study. Our main focus lies with the assessment

of correspondence among E-P data sets on a global and regional scale by the inter-comparison of six data sets and putting
the results into perspective regarding uncertainty estimates. Moreover, we investigate to what extent water budget closure is
achieved by satellite-based over-ocean estimates by comparing with ERA5 data and previously published estimates of water
cycle components.

Here, we consider the atmospheric water vapor budget with a focus on the oceans, where satellite observations of E are
available. The net change in atmospheric water vapor content can be written as:

$$\frac{\delta W}{\delta t} = E - P - \nabla \cdot (vq) \tag{1}$$

With $W$ the total column water vapor and $\nabla \cdot (vq)$ the moisture flux divergence, i.e., the amount of moisture removed by
dynamical transport from the considered volume. See Table 2 for all symbols and abbreviations. Compared to water vapor, the
contributions of liquid water and ice are very small (e.g., Berrisford et al. , 2011) and can be safely ignored in the context of
this study.

On the global scale $\nabla \cdot (vq)$ vanishes (as the Earth is a closed system) and Eq. 1 reduces to:

$$\Delta W = E - P \tag{2}$$

Where, for brevity, we write the $W$ tendency during large (monthly) time steps as $\Delta W$.

Assuming that $\Delta W$ is small compared to $E$ and $P$, Eq. 2 dictates that global total $E$ must equal global total $P$. Hence, an
observed imbalance in global totals of $E$ and $P$ indicates either an inconsistency in E and P data sets or a change in the global
water cycle, e.g. an increase in the amount of atmospheric water vapor (possibly caused by global warming), invalidating the
assumption that $\Delta W$ is negligible. Moreover, globally, $E$ and $P$ co-vary, meaning that their inter-annual, seasonal, and even
monthly variability are correlated.

At regional scales and for monthly averages, $\Delta W$ is small compared to $E - P$ and $\nabla \cdot (vq)$, so that Eq. 1 can be approximated
by:

$$E - P = \nabla \cdot (vq) \tag{3}$$

This is particularly valid for the large ocean and land regions, and since globally, $\nabla \cdot (vq) = 0$, from Eq. 3 it follows that:

$$(E - P)_{ocean} = \nabla \cdot (vq)_{ocean} = -\nabla \cdot (vq)_{land} = -(E - P)_{land} \tag{4}$$

with subscripts denoting summation over ocean or land. This separation into land and ocean contributions allows us to assess
the consistency of different E and P data sets, as satellite E data are not available over land.

In addition to the spatio-temporal distributions of individual budget terms, e.g., $E - P$, information on the accuracy and
precision of that value is of importance. Uncertainty estimates indicate whether observed differences — between data sets (e.g.,
observations and models), over time (trends, variability), or in space — are statistically relevant. Moreover, they play a major
role in data assimilation. Quantification of retrieval uncertainty, however, is a difficult task, particularly for non-linear retrieval
algorithms such as those used to retrieve $E$ and $P$ from satellite observations. Of the E CDRs investigated here, HOAPS-
4.0, OAFlux3, and SEAFLUX3 provide monthly mean uncertainty ranges. In HOAPS, random and systematic uncertainty

components are provided separately (Kinzel et al., 2016), allowing error propagation along with the calculation of temporal and/or spatial averages, as random errors (no covariance) disappear for large numbers of data points, whereas systematic errors (100% covariance) do not. For lack of information of error covariances, OAFlux3 and SEAFLUX3 monthly mean uncertainty estimates are similarly treated as having 100% covariance. An estimate of uncertainty is provided with ERA5 data in the form of results from a ten-member ensemble (Hersbach et al., 2020).

In the following section, we provide some background on $E$ and $P$ retrievals and introduce the E, P, and other data sets used for our study. Section 3 details the methods applied to the various data sets to enable a fair comparison. Results of our analyses are presented and discussed in Sections 4–5, and we close our study with a set of conclusions and recommendations.

## 2 Data sets

In this inter-comparison study, we assess the degree of agreement between five satellite-based E retrievals, two observation-based P retrievals, and a reanalysis data set. In this section, the retrieval algorithms will be briefly introduced: for more details, please refer to the literature listed in Table 1.

The retrieval of $E$ from satellite observations is challenging. It is determined from the bulk flux parameters near-surface wind speed and humidity gradient near the surface. Wind speed can be retrieved from satellite passive microwave brightness temperature (BT) measurements and BTs have also some sensitivity to near-surface specific humidity. Specific humidity at the ocean surface is derived from sea surface temperature (SST). All satellite-based E algorithms use reanalysis data to some extent and, *vice versa*, ERA5 also assimilates satellite data. Hence, these products cannot be considered completely independent and the distinction between "satellite data" and "reanalysis" is somewhat artificial and not always appropriate. However, for historical reasons — and for lack of a suitable alternative — we will retain these terms throughout this paper.

The main characteristics of the evaporation retrieval from passive microwave data are common to all satellite algorithms, but there is quite some variation regarding the input of Level-1 (calibrated observations) and Level-2 (retrieval results) data, as will be discussed below. First, we will give a brief description of the retrieval basics, followed by details of the various satellite algorithms.

### 2.1 Evaporation Data Records

The liquid-water equivalent evaporation rate, $E$, is calculated from the latent heat flux $Q_l$ as follows :

$$E = \frac{Q_l}{L_E} \quad (5)$$

Where $L_E$ is latent heat of evaporation of water. The latent heat flux, in turn, is parameterized according to the bulk flux algorithm (based on the Monin-Obhukov similarity theory representation of fluxes in terms of mean quantities):

$$Q_l = \rho L_E C_E u(q_s - q_a) \quad (6)$$

with $\rho$ the density of air, $C_E$ the coefficient of turbulent exchange, $u$ the wind speed at 10 m height relative to the ocean surface current speed, and $q_s$ and $q_a$ the specific humidity at the sea surface and at 10 m height, respectively. Whereas $q_a$ and $u$ are

derived from satellite observations of BT, $\rho$, $q_s$ and $L_E$ are derived from their dependences on SST and/or air temperature. The turbulent exchange coefficient $C_E$ is obtained from the Coupled Ocean-Atmosphere Response Experiment (COARE) version 3.0 algorithm (Fairall et al., 1996, 2003). The algorithm iteratively estimates stability-dependent scaling parameters and wind
gustiness to account for sub-scale variability.

Most of the data sets used here do not explicitly contain $E$, therefore, we calculated those from monthly means of $Q_l$ and SST using Eq. 5 and $L_E$ (in J/kg) given by (Henderson-Sellers, 1984):

$$L_E = 1.91846 \cdot 10^6 \cdot (\frac{T_s}{T_s - 33.91})^2 \tag{7}$$

where $T_s$ is SST in K. The slight difference with the definition of $L_E$ used in the COARE-3.0 algorithm causes negligible
differences of 0.03–0.04% for $T_s$ between 278–298 K.

The BT observations common to satellite-based retrievals of ocean turbulent fluxes come from the Special Microwave Imager (SSM/I, Hollinger et al., 1990) and Special Microwave Imager/Sounder (SSMIS, Kunkee et al., 2008) instruments on the Defense Meteorological Satellite Program (DMSP) platforms F08–F18. These data were corrected and inter-calibrated using various approaches to create FCDRs, stable fundamental climate data records (see, e.g., Wentz et al., 2013; Sapiano et
al., 2013; Berg et al., 2018; Fennig et al., 2020), which then serve as input to various satellite retrievals. Slight differences in calibration approaches lead to differences in FCDRs that propagate into the retrieved data. Issues with sensor stability, especially with SSM/I and SSMIS sensors, usually express themselves as slow drifts or sudden jumps of the global mean.

### 2.1.1 HOAPS-4.0

HOAPS (Andersson et al., 2010) relies almost completely on satellite data, as it only uses an ERA-interim profile climatology
as *a priori* starting point for the 1D-Var retrieval of $u$ and the humidity profile (Graw et al., 2017). The only other auxiliary data set is the daily Optimum Interpolated Sea Surface Temperature (OISST, Reynolds et al. (2007)), version 2, derived from AVHRR satellite data. OISST provides SST at a depth of 0.5 m which is transformed to a skin SST using the approach by Donlon et al. (2002), which is then used for the determination of $q_s$. The parameterization described in Bentamy et al. (2003) is used to determine $q_a$. For calculation of the flux parameters $Q_l$ and $E$, HOAPS-4.0 uses COARE version 2.6a (Bradley et
al., 2000), which is nearly identical to COARE-3.0 (Fairall et al., 2003). HOAPS-4.0 is a CDR derived from CM SAF (Climate Monitoring Satellite Application Facility) BT FCDR (Fennig et al., 2017, 2020) and is available at 0.5° and 6-hourly (except E-P) and monthly resolution from July 1987 — December 2014 (Andersson et al., 2017). HOAPS data can be obtained from https://wui.cmsaf.eu.

### 2.1.2 J-OFURO3

The latest update to J-OFURO involved improvements in the methods of flux retrieval and expansion of the data set in terms of time range and parameters (Tomita et al., 2019). The algorithm is similar to that described above. In addition to BT from SSM/I and SSMIS (from Remote Sensing Systems (RSS), Wentz et al., 2013), J-OFURO3 uses BT data from AMSR-E and AMSR2 (JAXA Version 3 and 2.1, respectively), and TMI (1B11 Version 7 from NASA-GESDISC) for the retrieval of flux parameters.

To determine $q_a$ a parameterization based on BTs, total column water vapor, and water vapor scale height was developed using match-ups of *in situ* buoy- and ship-based $q_a$ and DMSP-F13 BTs from eight channels (Tomita et al., 2018). From the instantaneous $q_a$ values, gridded daily averages are determined and inter-calibrated to DMSP-F13 $q_a$ to remove systematic differences caused by the use of different FCDRs. The $T_s$ required for the calculation of $q_s$ and other flux parameters is the median value of an ensemble of twelve *in situ*, satellite-based, and reanalysis data sets. Other auxiliary data sets include water vapor surface mixing ratios from ERA-interim (Dee et al., 2011), OSTIA sea ice concentration (Donlon et al., 2012), and air temperature from NCEP/DOE reanalysis (Kanamitsu et al., 2002). Near-surface wind speed is determined as the simple mean of values derived from microwave radiometers and scatterometers (Tomita et al., 2019). J-OFURO3 is available at $0.25°$ and daily resolution from 1988–2013. It was acquired from https://j-ofuro.scc.u-tokai.ac.jp/.

### 2.1.3 OAFlux3

Satellite data used for the production of OAFlux3 data include wind speed from active (scatterometer) and passive (radiometer) microwave instruments, SST from OISST (Reynolds et al., 2007), and $q_a$ from Goddard Satellite-Based Surface Turbulent Fluxes Dataset — Version 2; 2c (GSSTF2.0, Chou et al. (2003), Shie et al. (2009)). These are merged with NCEP and ERA40 reanalysis data using weighting factors that put more emphasis on satellite data (for $u$), on reanalyses ($q_a$), or weights both equally ($T_s$), whenever satellite data are available (Yu et al., 2008). OAFlux3 data are available from 1958–2018 (monthly) or 1985–2017 (daily) at $1°$ resolution from https://www.esrl.noaa.gov/psd/data/gridded/data.oaflux_v3.html .

### 2.1.4 IFREMER4.1

Similar to J-OFURO and OAFlux, IFREMER's ocean flux retrieval algorithm is based on a synergy of remote sensing and reanalysis data (Bentamy et al., 2013). The current version 4.1 contains, among others, latent heat flux (LHF) and SST at daily and monthly, $0.25°$ resolution from 1992–2018. The BTs used for retrievals are inter-calibrated by Colorado State University (CSU, Sapiano et al., 2013), except for data beyond June 2017, where CSU data ends and a switch to BTs from RSS (Wentz et al., 2013) is made. Inter-calibrated scatterometer wind data (Bentamy et al., 2017a) are supplemented by wind speeds determined by RSS from the SSM/I, SSMIS, and WindSat instruments. SST are from OISST (Reynolds et al., 2007). The model relating BTs to $q_a$ using satellite – *in situ* data match-ups was updated from Bentamy et al. (2003) and now includes two additional terms: $T_s$ and $T_a - T_s$ (with $T_a$ the air temperature at 10 m height from interpolated ERA-interim data (Bentamy et al., 2013)). IFREMER4.1 data were obtained via https://wwz.ifremer.fr/oceanheatflux/Data .

### 2.1.5 SEAFLUX3

The SEAFLUX3 data set consists of the near-surface meteorology and surface turbulent fluxes of heat, moisture, and momentum for the period 1988–2018 at an hourly, 25 km resolution (Roberts et al., 2020). An extension of the Roberts et al. (2010) neural network retrieval has been developed to estimate near-surface wind speed, humidity, and air temperatures from the GPM Level 1C intercalibrated BTs (Berg et al., 2018). Following the results of Roberts et al. (2019), the retrieval algorithms now

include additional *a priori* information on the vertical stratification of water vapor and lower tropospheric stability. A total of 14 passive microwave imagers including SSMI, SSMIS, TMI, AMSR-E, AMSR-2, and GMI are used for satellite retrievals and double differences are used to intercalibrate all estimates to the GPM GMI radiometer. The satellite retrievals are made in clear and cloudy scenes but are screened for precipitating conditions. A Kalman smoother is then applied to the retrieved estimates to blend the MERRA-2 (Modern-Era Retrospective analysis for Research and Applications, Version 2, (Gelaro et al., 2017)) background with satellite observations in an hourly gap-free analysis. A diurnally varying sea surface skin temperature from the SeaFlux-CDR (Clayson and Brown, 2016) is used together with the near-surface meteorology to estimate fluxes using the COARE 3.5 algorithm (Edson et al., 2013). Uncertainties are estimated for the individual near-surface meteorology as a blending of the retrieval and background errors through application of the Kalman smoother. Estimates of the surface flux uncertainties are computed using standard propagation of error techniques through the bulk flux algorithm.

### 2.1.6 ERA5

ERA5 is the current operational reanalysis running at ECMWF, the European Centre for Medium-range Weather Forecasts. Compared to its predecessor, ERA-interim, ERA5 includes improved model physics and data assimilation techniques, higher spatial (31 km) and temporal (1 hour) resolution. These lead to a gain in forecasting skill of up to one day compared to ERA-interim (Hersbach et al., 2020). Among many other observations, ERA5 assimilates CM SAF BT FCDR (Fennig et al., 2017); conditions for SST are prescribed using HadISST2.1. (Kennedy et al., 2016) and OSTIA (Donlon et al., 2012) from 09/2007 onwards (Hersbach et al., 2020). ERA5 encompasses data from ten reanalysis runs at a reduced spatial resolution of 62 km, allowing estimation of the uncertainty range from ensemble statistics. The analysis presented here is performed with the ECMWF ensemble mean, whereas uncertainty is determined from the ensemble. Both data sets were interpolated to $1°$ resolution at ECMWF.

The monthly averaged data set, available from the Copernicus Climate Data Store (https://cds.climate.copernicus.eu/), contains, among many others, total column water vapor (TCWV), vertically integrated moisture flux divergence (VIMD), total precipitation and evaporation rates (ECMWF, 2019). Monthly averages are calculated from daily means starting at 00 UTC and ending at 00 UTC the following day (ECMWF, 2020). Evaporation rates are derived from the gradients of specific humidity between the surface and the lowest model level (10 m for ERA5) as described above (ECMWF, 2016). The main differences between the satellite-based retrievals described here and ERA5 determination of $E$ are the consistency of atmospheric variables involved ($u$, $q_a$, $q_s$) and the high temporal sampling rate: monthly means are determined from (daily means of) hourly data from forecasts initialized daily at 6:00 and 18:00 UTC. Moreover, satellite-based data sets only provide fluxes over ocean, whereas ERA5 contains data over land and ocean. VIMD, i.e., the total amount of water vapor removed from the atmospheric column by dynamical transport, is provided in ERA5 as a gridded monthly mean field. We calculated the TCWV tendency in month $x$ from monthly mean ERA5 data by subtracting TCWV of month $x+1$ from TCWV of month $x-1$, then dividing by 30 days/month to obtain the mean TCWV tendency in km$^3$/day. This was converted to units of mm d$^{-1}$ by multiplication with the Earth's surface area for comparison with freshwater fluxes.

## 2.2 Precipitation Data Records

Microwave-based retrievals of precipitation are based on the interaction of liquid or solid hydrometeors with the upwelling radiation field. In HOAPS-4.0, $P$ is determined by an NN retrieval trained on profiles from an ERA-interim climatology (Andersson et al., 2010). The training data set consists of one month (August 2004) of assimilated SSM/I BTs and the corresponding ERA-interim $P$ (Bauer et al., 2006).

There is a multitude of global precipitation products in existence (see, e.g., Kidd and Huffman, 2011; Tapiador et al., 2017), but for this study we selected GPCP as the P data set with which to calculate E-P (except for the HOAPS product, which makes use of its own P data) because it is generally regarded as the data set that performs best globally. Moreover, J-OFURO also makes use of GPCP P to determine E-P (Tomita et al., 2019).

The Global Precipitation Climatology Project - 1 Degree Daily (GPCP-1DD; denoted GPCP hereafter), contains $P$ estimated from a combination of data from ground-based rain gauges and satellites — the latter including near-infrared, passive and active microwave observations (Huffman et al., 2001). Daily global precipitation rates are provided by GPCP-1DD at $1°$ resolution for the time range 1996–2017. We calculate monthly mean $P$ from version 1.3 GPCP-1DD (GPCP, 2018), because the spatial resolution of the monthly product is not sufficient for our purposes. These data were obtained from https://rda.ucar.edu/datasets/ds728.5/.

## 2.3 Errors, biases, uncertainty

Four out of seven data sets analyzed here contain explicit information on uncertainty. HOAPS contains estimates of random and systematic bias errors (Kinzel et al., 2016; Liman et al., 2018). The errors in $E$ were obtained by separating biases of HOAPS Level-2 $E$ with respect to collocated *in situ* ship-based data into equally populated $E$, $u$, $T_s$, and $W$ bins. The mean and standard deviation of the biases are assumed to represent the systematic and random components of the $2\sigma$ uncertainty range, respectively, which is probably a conservative estimate. By taking the approach of determining uncertainty ranges as a function of turbulent flux parameters these can also be assigned to times and regions not covered by the ship-based reference data set (Liman et al., 2018). For the current study, we calculated the mean uncertainty by averaging the systematic uncertainty component. The random component is negligible when averaging long time series. The HOAPS P data set does not contain uncertainty information; instead, a constant relative $1\sigma$ uncertainty range of $13\%$ was assumed, based on a comparison with ship-based *in situ* data (Burdanowitz, 2017). The total $E - P$ uncertainty was determined by error propagation.

Bias errors given in the OAFlux data set were computed based on the uncertainty ranges of individual input data sets, assuming no correlation between uncertainties from different data sets (Yu et al., 2008). Like for HOAPS uncertainty ranges, the OAFlux bias error was simply averaged for our investigations.

Uncertainties in SEAFLUX arise both from comparisons of the individual retrieval (e.g. wind speed, humidity, air temperature) errors evaluated against quality controlled buoy archives and that arising as a result of gap-filling through application of a Kalman smoother. Individual retrievals were generally found to be unbiased globally but some conditional biases likely remain. The total uncertainty is a measure of the reduction in retrieval uncertainties through combination of multiple sensors at

each location and time and increases in uncertainty related to sampling inhomogeneities. As the length of time grows between any given time and the previous or next observation, the sampling uncertainty increases. Thus the SEAFLUX uncertainties generally capture random retrieval uncertainties and sampling uncertainty but do not contain conditional systematic errors as developed for HOAPS. However, we note that the retrieval error itself does likely contain some components of conditional systematic biases even though the unconditional biases remain small.

In contrast to the monthly GPCP product, GPCP-1DD Version 1.3 does not provide explicit uncertainty estimates, hence here we assume a constant relative $1\sigma$ uncertainty range of $8\%$. This is the estimated bias error for GPCP data over the tropical oceans (Adler et al., 2012), which is where most of the P signal originates. Over the global oceans, the bias error was estimated at $10\%$, but Adler et al. (2012) considered this an upper bound.

In contrast to uncertainty ranges estimated by comparing with other (e.g., *in situ*) data sets, the uncertainty of ERA5 data is described by the standard deviation and the range of the ensemble, consisting of 10 seperate reanalysis runs (Hersbach et al., 2020). We determined these statistics after averaging of the data: first, the mean (e.g., global monthly mean) of each individual ensemble member was calculated, then standard deviation and range were determined. Note that ERA5 ensemble statistics should be interpreted in a relative sense (i.e., ensemble spread is larger where uncertainty is higher), as the numerical values are over-confident (ECMWF, 2020).

## 3 Methods

HOAPS is the only satellite data set containing E, P, and E-P data from a single source (i.e., microwave BTs). Within the HOAPS algorithm, $E - P$ is obtained by subtracting monthly mean $P$ from $E$ (Andersson et al., 2010). For this study, the data were remapped from $0.5°$ to $1°$. For the J-OFURO3 freshwater flux product, monthly mean GPCP-1DD $P$ are subtracted from the corresponding J-OFURO $E$ (Tomita et al., 2019). We determined $E - P$ of the other satellite-based data sets by subtracting monthly mean GPCP $P$ from the respective monthly mean $E$. These data sets will be denoted as IFREMER-G, SEAFLUX-G, and OAFlux-G to indicate that GPCP data were subtracted. J-OFURO, IFREMER and SEAFLUX do not provide $E$, therefore we calculated those from their respective LHF and SST data using Eqs. 5 and 7. The calculation of $E$ from $Q_l$ was performed at $0.5°$ and monthly resolution. Applying the same method of calculating E from HOAPS monthly mean LHF and SST data causes negligible differences with monthly mean E determined from instantaneous LHF and SST data (root mean square differences of $\leq 0.01$ mm d$^{-1}$ for individual grid boxes during 1997–2013). All E data were conservatively remapped to $1°$ to match GPCP resolution prior to subtraction of P. Similarly, ERA5 E-P was determined by subtracting monthly mean $P$ from $E$ at $1°$ resolution.

All comparisons presented here are performed with collocated data, i.e., only grid boxes (at $x$, $y$, and $t$) present in all data sets were used to create climatological or global averages. A more accurate collocation procedure would be performed at shorter, e.g., daily, time scale, because differences in filtering of high-precipitation scenes (where E retrieval is impaired) and selection of included satellite instruments lead to differences in sub-monthly sampling. This was, however, not feasible in this study, as HOAPS and J-OFURO E-P data are only provided on monthly resolution.

The satellite reference data set used in regional comparisons is determined by the statistical median of the satellite-based

data ensemble and therefore does not include ERA5. The median is chosen over the mean to exclude outliers. In the following, this reference data set is abbreviated SEM (satellite ensemble median).

Global averages were determined by converting the area-specific unit of mm d$^{-1}$ (equivalent to kg m$^{-2}$ d$^{-1}$) to units of (km$^3$ d$^{-1}$), computing the global (or ocean or land) mean and multiplying with the corresponding total surface area ($510 \cdot 10^6$, $350 \cdot 10^6$, and $160 \cdot 10^6$ km$^2$, respectively). Seasonally varying numbers of observations screened out due to sea ice are neglected.

Most comparisons in this study are shown in area-specific units, but for the comparison of global totals over land and ocean presented in Sect. 4.6, data were converted to area-integrated units (km$^3$ yr$^{-1}$) so that the totals balance.

Global total runoff from ERA5 and other data sets was determined by calculating the area integral of all points.

## 4   Results

### 4.1   Freshwater flux climatology

Freshwater flux climatologies obtained from 17 years of data (1997–2013) were determined from satellite ensemble median (SEM) and ERA5 data. They are shown in Fig. 1, panels a and b, to illustrate the overall spatial distribution of mean $E - P$. The chosen time range is the largest common time range of the data sets used in this study. Note that ERA5 data were matched to satellite data coverage.

Regions where mean $P > E$ are dominated by atmospheric freshwater outflux (into the ocean), shown in blue, and are

concentrated at the inter-tropical convergence zone (ITCZ) and the Pacific warm pool. In the subtropics, $E$ generally outweighs $P$. At higher latitudes $P$ and $E$ are approximately equal, but with a tendency to $E - P < 0$. Comparison of panels c-f with a and b shows that the E-P pattern is mainly determined by P in the tropical and high-latitude regions, but determined by E in the subtropical regions. The agreement between SEM and ERA5 E-P climatologies is good, yet, some systematic differences can be observed. Due to higher $P$ in the ITCZ, ERA5 shows more negative $E - P$ there. Conversely, the overall higher $E$ level

in ERA5 causes $E - P$ values larger than those found for SEM over most of the global oceans. Excessive E was also found to produce high E-P in ERA-interim (Brown and Kummerow, 2014).

The deviations are more apparent when climatological differences are analyzed. For this comparison we select ERA5 as a reference due to its spatio-temporal completeness and because it is the only "other" data set (i.e., not satellite data), keeping in mind that ERA5 data very likely also have inaccuracies and/or biases. Figure 2 shows climatological difference plots of

HOAPS (upper panel), OAFlux-G (middle panel), and SEAFLUX-G (lower panel) with collocated ERA5 data. Although HOAPS differences with ERA5 appear larger to the eye, root mean squared (RMS) differences are 0.6 mm d$^{-1}$ for each of the three comparisons: 0.60 mm d$^{-1}$ for HOAPS, 0.58 mm d$^{-1}$ for SEAFLUX-G, and 0.57 mm d$^{-1}$ for OAFlux-G. As already seen in Fig.1, differences are not homogeneously distributed over the globe. The HOAPS difference plot is characterized by an alternating pattern of positive and negative deviations. Stronger HOAPS $E$ in the subtropical central north and eastern South

Pacific produce elevated $E - P$ compared to ERA5. In contrast, elevated ERA5 $E$ over the east China Sea combines with smaller ERA5 $P$ in the region, resulting in higher ERA5 $E - P$. The positive bands on either side of the equator are due to

higher HOAPS $E$, whereas the negative $E - P$ differences at the equator are due to smaller HOAPS $P$. The negative deviations to the east and west of Australia are also due to differences in $P$, whereas the deviations at latitudes $> 40°$ S are due in equal parts to $E$ and $P$. The differences between OAFlux-G and ERA5 are mainly due to $P$, apart from the regions in the subtropical Pacific and Atlantic Oceans, where OAFlux $E$ is smaller than ERA5 $E$. SEAFLUX-G shows slightly larger differences with ERA5. In the band within $30°$ of the equator, SEAFLUX yields higher $E$ than ERA5 (and OAFlux) in most of the Pacific and Atlantic Ocean, except in the upwelling regions on the west coasts of Africa and the Americas. The difference plots of J-OFURO and IFREMER-G with ERA5 are not shown here, but are very similar to the lower left panel because the differences in $P$ between GPCP and ERA5 are larger than differences in $E$ in most regions. All plots, including difference climatologies of $E$ and $P$, can be found in the Appendix, Fig. A1.

To investigate where the differences are significant, the right column of Fig. 2 presents the $1\sigma$ uncertainty range from HOAPS (upper panel), OAFlux-G (middle panel), and SEAFLUX-G (lower panel). Moreover, regions where the difference between satellite $E - P$ and ERA5 $E - P$ are greater than the $2\sigma$ uncertainty range are enclosed by white contour lines in the left panels. The ERA5 E-P uncertainty shows a pattern similar to that of OAFlux-G, but is a factor of $10$ smaller than the uncertainties estimated for satellite data and therefore adds a negligible component to the total uncertainty estimate. The HOAPS uncertainty range is larger than HOAPS-ERA5 $E - P$ differences over most of the globe. This is mainly due to $P$, for which we assumed $13\%$ uncertainty. The deviations $> 1$ mm d$^{-1}$ in the oceans' desert regions (off the west coasts of Peru and Southern Africa) and in the higher latitudes are clearly outside the $2\sigma$ uncertainty ranges. In contrast, OAFlux-G $E - P$ deviations are larger than the estimated $2\sigma$ uncertainties in the ITCZ, the west coasts of the Pacific and Atlantic Ocean, the Arabian Sea, and the Southern high latitudes. Again, the uncertainty range is mainly given by $P$, for which we assumed a relative uncertainty of $8\%$. Due to the small uncertainty estimates in SEAFLUX, all of the larger differences with ERA5 in the Atlantic and Pacific Oceans are significant.

### 4.2 Inter-comparison of freshwater flux over ocean: global means

Monthly mean $E$, $P$, and $E - P$ of six (or three) data sets were collocated (see Section 3) and averaged over the global oceans ($80°$S – $80°$N). Climatological seasonal cycles were determined for the overlapping time range (1997–2013) and are shown in Fig. 3, panels a–c. HOAPS, ERA5, OAFlux, SEAFLUX, and GPCP uncertainty ranges are presented in the boxes attached to the right of panels a–c. Dots show the climatological mean value and error bars indicate the associated $1\sigma$ uncertainty Subtracting the seasonal cycle from the respective monthly mean time series yields global ocean anomalies of $E - P$, $E$, and $P$, which are presented as three-month running means in panels d–f. Seasonal and inter-annual variability are of the same order of magnitude, which can be seen by comparing the left panels with those on the right (the y-axis spans $1$ mm d$^{-1}$ in all panels).

There are substantial deviations between $E$, $P$, and $E - P$ data. Panel a shows that a difference of about $0.2$ mm d$^{-1}$ is found between OAFlux and J-OFURO $E$. An additional discrepancy of $0.2$ mm d$^{-1}$ exists between J-OFURO and ERA5. E data from HOAPS, IFREMER, and OAFlux are much closer to each other: satellite-based $E$ all fall within the OAFlux uncertainty range (red error bars), whereas the ERA5 climatological mean $E$ does not fall within the larger HOAPS uncertainty range. The

HOAPS uncertainty range is much larger than the seasonal variation, which indicates that it is likely overestimated, which may
be due to the assumption of $100\%$ covariance for systematic uncertainty.

Panel b shows that the seasonal cycle of global ocean mean $P$ is shallow and the two satellite-based data sets agree within
the GPCP uncertainty for ten months of the year. Like for $E$, we find substantial differences among the three P data sets: there
is a deviation of about $-0.1$ mm d$^{-1}$ between HOAPS and GPCP, and ERA5 shows values that are about $0.25$ mm d$^{-1}$ higher
than GPCP, which was also found by Hersbach et al. (2020). These differences can, in part, be explained by differences in P
frequency distributions and, in particular, by the fraction of rain occurrences, which is much lower in HOAPS than in GPCP
or ERA5. This will be discussed in Sect. 5. Since in this paper the focus is on the inter-comparison of E-P (not specific E or
P algorithm issues), we only describe the observed differences between P (and E) data sets to obtain a better understanding of
differences between E-P data.

Apart from HOAPS $E - P$ in March–April, all satellite data sets agree on phase and amplitude of the $E - P$ seasonal cycle
(panel c). ERA5 shows hardly any dependence on season, as the magnitude of the summer maximum is smaller in ERA5 due
to the relatively larger summer $P$ maximum. The monthly and inter-annual variability of ERA5 $E - P$ is, like the seasonal
cycle, of smaller amplitude than that of satellite data which is caused by the high degree of coherence between $E$ and $P$, and
will be discussed in more detail in Sect. 4.5. Because compared to satellite data, ERA5 $E$ and $P$ are biased high by about the
same amount, $E - P$ is close to the satellite data. HOAPS yields the highest $E - P$ due to its low mean $P$. All E-P data are
contained within the HOAPS and OAFlux uncertainty ranges.

The $E$ anomalies in panel d display a high degree of correlation on the monthly time scale. On the multi-annual scale all
data sets show some degree of variability, which is most likely linked to sensor and inter-calibration issues (e.g., Robertson
et al., 2020), and the variability is not consistent. For example, the slow, decadal-scale oscillation observed in HOAPS and
IFREMER appears to be in anti-phase compared to OAFlux. The three P data sets yield inter-annual variations with amplitudes
that are similar in amplitude to those found for E, and show a high degree of correspondence in their monthly and inter-annual
variability — apart from the stronger dependence of HOAPS on ENSO (El Niño - Southern Oscillation) phase. This is a known
characteristic of HOAPS data (see, e.g., Andersson et al., 2011; Masunaga et al., 2019), and is most apparent in panel e, where
the Niño 3.4 SST index (Trenberth and Stepaniak, 2001) is plotted in gray bars along with $P$ anomalies: HOAPS $P$ correlates
with Niño 3.4 if a lag of 3 months is taken into account ($R^2 = 0.73$). Apparent agreement is found among all $E - P$ anomalies
(panel f) — again apart from the ENSO-related deviations found in HOAPS P. The agreement among $E - P$ anomalies is best
in the "quiet" ENSO years (2001–2005), but this is probably a coincidence as the spread in $E - P$ in other years is mainly due
to differences in $E$ and not in $P$. Note that differences between J-OFURO-G, IFREMER-G, SEAFLUX-G, and OAFlux-G are
due to differences in $E$, as in all cases GPCP $P$ was used for the calculation of $E - P$.

**4.3 Inter-comparison of freshwater flux over ocean: time series on regional scales**

In this section, we investigate the temporal correlation of water cycle components on regional scales. This approach will help
to understand differences between the various data sets by uncovering in which regions the differences are particularly large
(or small). As a reference for the E and E-P comparisons, we use SEM, a data set determined by the statistical median of

all satellite data sets. Since we use only two satellite P data sets, GPCP is selected as a reference for the P comparison. We

determine correlation coefficient, slope and intercept of the linear regression ($y = ax + b$) between $1° \times 1°$ monthly means (not anomalies) of each data set, $y$, and the reference, $x$, to examine where estimates are most consistent.

The results are shown for all six E data sets in Fig. 4, where the left column displays the correlation coefficient. On the top row, HOAPS yields $R^2 > 0.75$ over most of the globe, with some notable exceptions in the ITCZ and the Peruvian coast. The other satellite data yield higher correlation coefficients. The correlation pattern of ERA5 with SEM is similar to that found for

HOAPS, although the tropical areas with $R^2 < 0.75$ are not at the same locations. The highest overall correlation with SEM is found for J-OFURO and SEAFLUX, with $R^2$ exceeding $0.75$ essentially everywhere.

The middle panels of Fig. 4 display the slope of the linear regression. A slope greater (smaller) than 1 implies an over- (under-) estimation, particularly of large values, compared to SEM. HOAPS overestimates $E$ in the tropics, except in an area in the eastern Pacific at $0° - 5°$ N, where $a < 1$. J-OFURO, IFREMER, and OAFlux each yield slopes $< 1$ within $30°$ of the

equator and slopes close to 1 everywhere else (apart from the band with $a < 1$ seen in IFREMER at high southern latitudes). Of those three E data sets, OAFlux displays the largest deviations from $a = 1$. In contrast, SEAFLUX yields slopes close to unity over the whole globe. An inhomogeneous pattern is found for ERA5, but the slope is generally close to 1. A small region in the tropical Atlantic stands out due to its large slope, and since this is not seen in any of the satellite data sets, it must be a feature in ERA5 data.

The patterns in the middle panels are nearly all mirrored in the right panels, i.e., wherever large values are overestimated ($a > 1$), small values are underestimated ($b < 0$), and *vice versa*. All data sets thus appear to agree on intermediate values. Overall, the correspondence between E data sets is best in the subtropics, while the largest deviations appear mainly in the tropics. This is due to the frequent occurrence of weather conditions in which the moisture stratification departs substantially from typical conditions to which the retrieval algorithms of near-surface moisture are tuned. Accounting for this dependence

on moisture stratification, as in the SEAFLUX and J-OFURO algorithms, improves retrieval results appreciably compared to *in situ* measurements (Roberts et al., 2019).

Figure 5 shows the same analysis for $P$ from HOAPS (upper panels) and ERA5 (lower panels). The correlation coefficient between HOAPS and GPCP $P$ is $> 0.75$ in the ITCZ and about $0.5$ for most of the global oceans. In the oceans' deserts $R^2 < 0.25$ are found, which is mostly due to the small dynamic range of mean $P$. Compared to GPCP, HOAPS underestimates

$P$ in this region, as $a < 1$. At latitudes poleward of $50°$ similarly small $R^2$ are found that are in part due to the small dynamic range, and in part to difficulties pertaining to the detection of snow by passive microwave instruments (Tapiador et al., 2017; Kidd and Huffman, 2011). HOAPS underestimates high $P$ here and overestimates small $P$ ($b > 0$ mm d$^{-1}$) compared to GPCP. Very similar patterns are seen for ERA5, although in general, the correlation coefficient is higher than for HOAPS. ERA5 is biased high almost everywhere compared to GPCP. Both HOAPS and ERA5 show a smaller range of P in the Southern Oceans,

as the slope is less than $0.5$, but the large intercept indicates an overestimation of small P compared to GPCP. The narrow band of $R^2 < 0.75$ and $b > 1$ mm d$^{-1}$ at the equator is also found in both HOAPS and ERA5.

For E-P, the results of the regression analysis are shown in Fig. 6. The highest correlation coefficients (and slopes and intercepts closest to 1 and 0 mm d$^{-1}$) are found among the data sets calculated with GPCP P. This shows that most of the

variability in E-P is due to differences in P. Since GPCP P is used in four out of five data sets included in the SEM, those data sets show high correlations, whereas HOAPS and ERA5 yield patterns very similar to those found for the P comparison in Fig. 5. Nevertheless, both for ERA5 and HOAPS the correlation in most of the tropics is higher for E-P than for P. In summary, the correlation patterns for HOAPS and ERA5 indicate agreement on the seasonal cycle in the tropics, a result found previously by Brown and Kummerow (2014), although we find that its amplitude is reduced in the GPCP-based E-P data (Fig. 3). Less agreement is found in the Southern Oceans, where GPCP-based $E-P$ are underestimated relative to SEM. In the mid-latitudes, the regression with SEM yields slopes near 1 and intercepts close to 0 mm d$^{-1}$ for ERA5 and HOAPS, but the correlation is less than in the tropics, probably due to the smaller dynamic range of E-P.

In the present study, we compare satellite-based E-P with ERA5 E-P because we are also examine the separate contributions from E and P. It can, however, be argued that VIMD from reanalysis is a more reliable quantity than reanalysis E-P, since VIMD is calculated from the state variables wind and water vapor, whereas E and P are model-physics derived (e.g., Trenberth et al., 2011)). We verified that in ERA5, the agreement between $E-P$ and $\nabla \cdot (vq)$ is generally good, as shown in Appendix B. Hence, changes to the plots in Fig. 6 are minor when ERA5 $\nabla \cdot (vq)$ is used to calculate the regression with SEM $E-P$ in stead of ERA5 $E-P$, as shown in Fig. A2 in the Appendix.

## 4.4 Examination of the water budget in ERA5

One way of investigating the consistency of different water cycle components is determining if the global water budget (Eq. 1) is closed. However, satellite E-P data sets are available over ocean only, so we revert to a comparison with gap-free reanalysis data. There is no internal constraint for budget closure in ERA reanalyses (Berrisford et al. , 2011; Hersbach et al., 2020), and as the budget was not closed in ERA-interim, it is worthwhile to investigate ERA5's behavior in this regard. Monthly mean total ERA5 $E-P$ over the globe, the ocean, and land are shown in Fig. 7 in black, blue, and green, respectively. The mean values over the globe and land were scaled by their surface area relative to the ocean surface area (i.e., they were multiplied by $510/350$ and $160/350$, respectively) to obtain consistency with the over-ocean means shown in Fig. 3. The error bars on ERA5 data depict the standard deviation of the 10-member ensemble. Nearly all of the uncertainty in global mean E-P is due to the uncertainty over ocean; the error bars on the over-land E-P are smaller than the graph's line width. This is due in equal parts to E and P, which have similar ensemble standard deviations (not shown). For the time range shown in Fig. 7 global $E-P$ is seen to oscillate around 0 mm d$^{-1}$, meaning that the ERA5 water budget is closed on a yearly time scale (in agreement with the findings by Hersbach et al., 2020). The seasonal cycle is mainly driven by increased evapo-transpiration of vegetation on land and peaks in northern hemispheric summer due to the larger fraction of land in the Northern Hemisphere. Precipitation shows a similar seasonal cycle over land, but does not completely cancel out in E-P due to a slight phase shift with respect to the E seasonal cycle (not shown).

Figure 7 shows that monthly means of global $E-P$ and $\Delta W$ (light blue line) display a high degree of coherence, as expected from Eq. 2. This is an indication that the (atmospheric) water cycle is well represented in ERA5.

Globally, VIMD is zero, as no water vapor is transported out of (or into) the Earth system. However, we find ERA5 global total VIMD to be $-0.04$ mm d$^{-1}$: a small value within the standard deviation of the ensemble of single grid boxes, but

significant and on the order of the amplitude of the seasonal cycle of net E-P on the global scale. The deviation from zero is due to the fact that VIMD is calculated in grid point space (and not in the model's spectral space), where the mathematical constraint of net zero divergence is not enforced (P. Berrisford, personal communication, Oct. 2020). Interestingly, VIMD over land (pink) agrees well with over-land E-P, whereas VIMD over ocean (purple line) is smaller than over-ocean E-P also by $-0.04$ mm d$^{-1}$. Based on the results of the regression analysis shown in the upper panels of Fig. A2 we speculate that discrepancies between $E - P$ and $\nabla \cdot (vq)$ over the ocean's desert regions also play a role in causing $\nabla \cdot (vq)$ to be smaller over ocean than over land.

## 4.5 Examination of the water budget in satellite data sets

Globally, $E - P$ is equal to $\Delta W$ (Eq. 2 and Fig. 7), and because $\Delta W$ is two orders of magnitude smaller than $E$ and $P$, global mean $E$ should necessarily be almost equal to global mean $P$. This seemingly trivial finding provides us with a tool to investigate the consistency of E and P data sets: by determining how well they correlate. For ERA5, global mean $E$ and $P$ yield correlation coefficients $R^2 = 0.82$ and $R^2 = 0.84$ for monthly and yearly means, respectively. This procedure cannot be applied to the satellite E-P data considered here, as they contain values over ocean only. Since there is a substantial seasonality in water vapor transport (Fig. 7), the correlation between ocean-only $E$ and $P$ is expected to be much lower. A regression of ERA5 $E_{ocean}$ and $P_{ocean}$ monthly means (where the subscript $_{ocean}$ indicates averaging over ocean only) indeed yields only $R^2 = 0.42$, or $R^2 = 0.57$ for yearly means. To account for the net transport of water from ocean to land, we include $\nabla \cdot (vq)_{ocean}$ into the analysis, and, applying Eq. 4, correlate $E_{ocean} - \nabla \cdot (vq)_{ocean}$ with $P_{ocean}$. For ERA5, the resulting $R^2$ are $0.86$ for yearly and monthly means: very similar to the coefficients found for the correlation of global $E$ with $P$.

We calculated correlation coefficients of the various E and P data sets used in this study, combined with ERA5 $\nabla \cdot (vq)_{ocean}$, and listed them in Table 3. The analyses were performed separately on (i) monthly means, primarily indicating agreement on the seasonal cycle; (ii) yearly means, a measure of consistency of inter-annual variability, including trends; and (iii) monthly anomalies, focused on short-scale variability.

The small correlation coefficients found for monthly mean satellite data in part reflect the differences in the seasonal cycles of E and P (see panels a and b of Fig. 3). But the results from the analysis of monthly anomalies (where the mean seasonal cycle was subtracted) are very similar to those found for monthly means, indicating that, compared to monthly and inter-annual variability, the seasonal cycle is of lesser importance on the global scale. Including the contribution of $\nabla \cdot (vq)_{ocean}$ improves the correlation appreciably for ERA5, as mentioned above. For satellite data the correlation also improves, particularly for yearly means and monthly anomalies of IFREMER-G and J-OFURO-G (not shown). On a yearly time scale, we do not expect a high degree of correlation, as inter-annual variability is small and no clear trends are observed in panels d and e of Fig. 3. For ERA5, $R^2 = 0.86$, but this is primarily caused by small $E$ and $P$ in 1997 and 1999, which is also the case for IFREMER. The correlation found for J-OFURO, $R^2 = 0.31$, is the highest found among satellite data. The remaining satellite data sets are not significantly correlated on a yearly time scale (p-value $> 0.05$). Clearly, time series longer than the 17 years investigated here would benefit the analysis of yearly mean data.

Overall, this analysis shows that satellite-based estimates of E are less consistent with satellite-based P data than ERA5 E and P. To a certain degree, this is expected, as the three variables used in the analysis come from different sources (e.g., E from IFREMER, P from GPCP, and $\nabla \cdot (vq)_{ocean}$ from ERA5), each with its own sampling and uncertainty characteristics. Nevertheless, from the global water cycle perspective, some degree of correspondence between $E_{ocean}$ and $P_{ocean}$ is expected.

## 4.6  Estimates of global total water cycle components

From the data shown in the previous sections we calculated climatological (1997–2013) global ocean total $E$ and $P$, total $E - P$ (separated into land and ocean contributions), runoff and net transport. The latter is equal to the total over-ocean (or over-land) $\nabla \cdot (vq)$. Our results are given in the three upper rows of Table 4. For brevity, we again denote total $E$, $P$, and $E - P$ over ocean as $E_{ocean}$, $P_{ocean}$, and $(E - P)_{ocean}$ respectively, and similarly, over-land $E - P$ as $(E - P)_{land}$. For better comparison with earlier estimates, values are given in units of $10^3$ km$^3$ yr$^{-1}$.

The rows labeled "obs." display the mean, standard deviation (std) and range of $P_{ocean}$ (from GPCP and HOAPS) and $E_{ocean}$ (from HOAPS, J-OFURO, IFREMER, SEAFLUX, and OAFlux). Various estimates of global totals of water cycle components can be found in the literature, for example in (Oki and Kanae , 2006; Trenberth and Asrar , 2014; Rodell et al., 2015; Allan et al., 2020), and are shown in the lower half of Table 4.

The largest spread among water cycle components is found for E and P over ocean, both in an absolute and a relative sense, each with a range spanning about $100 \cdot 10^3$ km$^3$ yr$^{-1}$, or 20–25%. The relative spread in $R$ is similar, but is a factor of ten smaller in absolute values. In their study, Rodell et al. (2015) estimated $E_{ocean}$ from observations at $410 \cdot 10^3$ km$^3$ yr$^{-1}$ (corresponding to 3.21 mm d$^{-1}$), in agreement with Trenberth and Asrar  (2014), but found a value of $450 \cdot 10^3$ km$^3$ yr$^{-1}$ by applying an algorithm that optimized all water cycle components to achieve water and energy budget closure. The algorithm caused a concurrent increase in $P_{ocean}$ from the observed $385 \cdot 10^3$ km$^3$ yr$^{-1}$ to $403 \cdot 10^3$ km$^3$ yr$^{-1}$. The global total fluxes estimated by Allan et al. (2020) derive from Rodell et al. (2015), but following the recommendation by Stephens et al. (2012), $E_{ocean}$ and $P_{ocean}$ were both increased by $30 \cdot 10^3$ km$^3$ yr$^{-1}$ to improve the agreement with energy constraints, yet keeping land-ocean fluxes constant. These increases are larger than the $\pm 22 \cdot 10^3$ km$^3$ yr$^{-1}$ uncertainty on $E_{ocean}$ and $P_{ocean}$ estimated by Rodell et al. (2015) based on the optimized method and so a more modest increase of about $20 \cdot 10^3$ km$^3$ yr$^{-1}$ may be more appropriate. These would produce fluxes of $E_{ocean} = 470 \cdot 10^3$ km$^3$ yr$^{-1}$ and $P_{ocean} = 424 \cdot 10^3$ km$^3$ yr$^{-1}$ that are quite close to ERA5 estimates (R. Allan, personal communication, Oct. 2020).

In our study, we find a large range of $E_{ocean}$: HOAPS yields $397 \pm 96 \cdot 10^3$ km$^3$ yr$^{-1}$, OAFlux $414 \pm 37 \cdot 10^3$ km$^3$ yr$^{-1}$, IFREMER $418 \cdot 10^3$ km$^3$ yr$^{-1}$, SEAFLUX $444 \pm 5 \cdot 10^3$ km$^3$ yr$^{-1}$, and J-OFURO $453 \cdot 10^3$ km$^3$ yr$^{-1}$. Note that HOAPS $1\sigma$ uncertainty is as large as the range among satellite-based $E_{ocean}$ and more than three times the corresponding std, again imply-ing an overestimation of the HOAPS uncertainty range (see Sect. 4.2). The OAFlux $1\sigma$ uncertainty is of the same magnitude as the std among satellite-based $E_{ocean}$, whereas the SEAFLUX uncertainty estimate is small in comparison. The small $E_{ocean}$ found by HOAPS is partly due to data coverage, as data are only available over the ice-free ocean within $80°$ of the equator. A test with ERA5 data showed that $E_{ocean}$ decreases by $5\%$ when the data are adapted to HOAPS coverage. Conversely assuming

a 5% increase for HOAPS yields $417 \cdot 10^3$ km$^3$ yr$^{-1}$. The same reasoning applies to the other satellite data sets with similar effects on $E_{ocean}$.

The spread in $P_{ocean}$ is of the same magnitude as that found for $E_{ocean}$: HOAPS yields $335 \pm 44 \cdot 10^3$ km$^3$ yr$^{-1}$, GPCP $384 \pm 31 \cdot 10^3$ km$^3$ yr$^{-1}$, assuming uncertainty ranges of 13% and 8% for HOAPS and GPCP, respectively. ERA5 yields $426 \pm 2 \cdot 10^3$ km$^3$ yr$^{-1}$, which is significantly larger than either HOAPS or GPCP. From the GPCP-1DD data used in this study, we determine $P_{land} = 116 \cdot 10^3$ km$^3$ yr$^{-1}$, which is close to the estimates presented previously (which range from $111 \cdot 10^3$ km$^3$ yr$^{-1}$ (Oki and Kanae , 2006) to $117 \cdot 10^3$ km$^3$ yr$^{-1}$ (Rodell et al., 2015)). This is due to the fact that GPCP is used for all observation-based estimates. ERA5 $P_{land}$ is somewhat higher than the observations ($122 \cdot 10^3$ km$^3$ yr$^{-1}$), but the difference is not significant.

From the estimates of $E_{ocean}$ and $P_{ocean}$ it follows that for HOAPS $(E-P)_{ocean} = 65 \pm 106 \cdot 10^3$ km$^3$ yr$^{-1}$, for OAFlux $(E-P)_{ocean} = 35 \pm 48 \cdot 10^3$ km$^3$ yr$^{-1}$. IFREMER yields $(E-P)_{ocean} = 38 \cdot 10^3$ km$^3$ yr$^{-1}$, J-OFURO $(E-P)_{ocean} = 59 \cdot 10^3$ km$^3$ yr$^{-1}$, and SEAFLUX $(E-P)_{ocean} = 61 \pm 31 \cdot 10^3$ km$^3$ yr$^{-1}$.

The spread in $R$ ($40$–$51 \cdot 10^3$ km$^3$ yr$^{-1}$) is quite large. The total continental runoff from ERA5 is $41.2 \cdot 10^3$ km$^3$ yr$^{-1}$, which is slightly higher than the $39 \cdot 10^3$ km$^3$ yr$^{-1}$ found in ERA-interim (Berrisford et al. , 2011). Data from the Global Runoff Data Centre (GRDC, Wilkinson et al., 2014) yield an average of $41 \cdot 10^3$ km$^3$ yr$^{-1}$ with a standard deviation of $1.8 \cdot 10^3$ km$^3$ yr$^{-1}$ for 1987–2010. They are at the lower bound of the estimates by Clark et al. (2015), who find $44.2 \pm 2.7 \cdot 10^3$ km$^3$ yr$^{-1}$ for 1950–2008. The same study cites estimates by various authors that range from $25$–$50 \cdot 10^3$ km$^3$ yr$^{-1}$, with those based on freshwater fluxes representing the lower boundary ($25$–$39 \cdot 10^3$ km$^3$ yr$^{-1}$). The long-term average runoff estimated from the GRUN (Global RUNoff, Ghiggi et al., 2019) data set is $38 \cdot 10^3$ km$^3$ yr$^{-1}$, consistent with the above-mentioned range, albeit somewhat smaller than the best estimate by Clark et al. (2015). Note that GRUN runoff estimates are not independent of reanalysis data, as the machine-learning algorithm uses surface temperature and P data from 20CR reanalysis (Compo et al., 2011; Ghiggi et al., 2019). Improvements in the quality of E-P estimates will aid the quantification of river runoff by providing an independent estimate of the total freshwater flux.

Where runoff is the net transport of (liquid) water from land to ocean, over-ocean VIMD ($\nabla \cdot (vq)_{ocean}$) is, to a good approximation, the net amount of water vapor advected from ocean to land. Hence, $R = \nabla \cdot (vq)_{ocean} = -\nabla \cdot (vq)_{land}$. Whereas ERA5 estimates of $\nabla \cdot (vq)_{land}$ are at the high end of the range of $R$ mentioned above, at $31 - 33 \cdot 10^3$ km$^3$ yr$^{-1}$, $\nabla \cdot (vq)_{ocean}$ is too small. As observed above, the fact that ERA5 VIMD is calculated in grid point space causes global total $\nabla \cdot (vq)$ to be about $10 \cdot 10^3$ km$^3$ yr$^{-1}$, and not zero. In addition, due to the tighter observational control over land, analysis increments may be larger over ocean than over land and may cause net $\nabla \cdot (vq)$ to be close to net $E - P$ over land, but less so over ocean (P. Berrisford, personal communication, Oct. 2020). There is another field in the ERA5 archive, the vertical integral of divergence of moisture flux (VIWVD, parameter ID p84.162), which is very similar to VIMD but is computed from hourly instantaneous reanalysis fields (and contains no contributions from liquid or solid water — but these can be neglected for our purposes). Globally VIWVD adds up to $0.9 \cdot 10^3$ km$^3$ yr$^{-1}$ (0.003 mm d$^{-1]}$), a factor of 10 smaller than total VIMD. In addition, the agreement between over-ocean VIWVD and $(E-P)_{ocean}$ is much better than that found for VIMD and $(E-P)_{ocean}$, and at $41.6 \cdot 10^3$ km$^3$ yr$^{-1}$ and $-40.7 \cdot 10^3$ km$^3$ yr$^{-1}$, respectively, over-ocean VIWVD and over-land VIWVD are also in agreement

with other values in the five right-most columns of Table 4. The estimates of net transport by Oki and Kanae (2006); Trenberth and Asrar (2014); Rodell et al. (2015) are in agreement with ERA5 $R$ and $\nabla \cdot (vq)_{land}$. The consistency between runoff and net transport seen in the last four rows of Table 4 is mainly by construction, as both are usually required (or defined) to be equal.

The five right-most columns of Table 4 should, theoretically, all contain identical values (except for the sign). In practice, however, $(E - P)_{land}$ ranges from $-46$ to $-40 \cdot 10^3$ km$^3$ yr$^{-1}$, $(E - P)_{ocean}$ from 24 to $52 \cdot 10^3$ km$^3$ yr$^{-1}$, $\nabla \cdot (vq)$ ranges from 40 to $51 \cdot 10^3$ km$^3$ yr$^{-1}$ (disregarding the erroneous ERA5 $\nabla \cdot (vq)_{ocean}$ value), and $R$ from $40$–$51 \cdot 10^3$ km$^3$ yr$^{-1}$. Assuming the degree of consistency found among these values represents the reliability of the estimate, it is clear that E-P uncertainty is largest over ocean, and from the first two columns of Table 4 it follows that E and P contribute almost equally to that uncertainty.

## 5  Discussion

We present an inter-comparison of five recent satellite-based E-P data sets. All five E data sets are the latest official versions of CDRs generated from (different) BT FCDRs and are combined with GPCP (or HOAPS) P CDR to form E-P data.

Although it is tempting to make a ranking from the results of our inter-comparison, there are good reasons to resist. First, there are not enough truly independent data with which to assess the quality of each data set. And second, each data set has its particular strengths and weaknesses: for example, HOAPS comes closer to water budget closure than OAFLUX or IFREMER (panel c of Fig. 3).

Of the E data sets used in this study, HOAPS depends on the least amount of model data, using these on a climatological (as opposed to collocated, instantaneous) basis. ERA5, being a reanalysis, represents the other extreme, and the remaining retrieval algorithms are somewhere in between. All algorithms, including ERA5 physics, rely on the same parameterization of bulk fluxes (Eq. 6) and on the COARE algorithm for the determination of the turbulent exchange coefficient (see Sect. 2). The origin of $E$ differences between various data sets must therefore lie with the bulk flux parameters $u$, $q_a$, and $q_s$, and with differences in sampling characteristics. A recent study by Roberts et al. (2019) showed that HOAPS, SEAFLUX, and J-OFURO retrieve global mean $q_a$ that are systematically too small compared to *in situ* ship-based NOCSv2 data (Berry and Kent, 2011); IFREMER slightly overestimates $q_a$. This difference could be largely improved by applying a correction based on the subtraction of a regime-dependent bias (in which regimes are defined by their water vapor vertical stratification, cloud liquid water content, and SST; and the bias determined with respect to NOCSv2). The HOAPS algorithm determines $Q_l$ (and $E$) systematic error estimates in a similar fashion: biases with respect to ship-based data were binned by $Q_l$, $u$, $T_s$, and $W$, then collected into a 4-dimensional look-up-table (Kinzel et al., 2016; Liman et al., 2018). Subtracting the systematic error from HOAPS $Q_l$ (or $E$) would raise the global mean and improve the agreement with ship-borne data sets such as NOCSv2. Initial tests show that this is, indeed, the case for $Q_l$, and is the topic of a forthcoming study. We stress, however, that reducing biases with respect to reference (e.g., *in situ*) data by improving the retrieval algorithm through better understanding of physical processes should be the preferred way forward.

Forcing improved agreement of satellite-based estimates of $E$ with respect to *in situ* data (and ERA5) via bias adjustment has a downside: the bias removed from $E$ reappears in $E - P$, which is now in agreement with different estimates of $(E - P)_{land}$, $\nabla \cdot (vq)$, and estimates of continental runoff rates. Satellite $E - P$ is also in agreement with ERA5 $E - P$ due to the cancellation of differences, which was already noted in Sect. 4, when discussing the large positive biases of ERA5 $E$ and $P$ with respect to satellite observations (Fig. 3, b and c). It is interesting to note that satellite-based E are very likely biased high by the removal of scenes with strong precipitation (where the retrieval of wind speed, LHF, and E is not possible). In this light, the difference in $E$ between ERA5 and the satellite-based retrievals should actually be larger than observed in Fig. 3, as monthly mean $E$ is determined from all sky conditions in reanalysis. As OAFlux and SEAFLUX blend satellite estimates with continuous background fields (Sect. 3), these algorithms should be less impacted by such sampling biases.

Andersson et al. (2011) found a high bias of HOAPS-3.2 E-P for the time period 1992–2005, which was much reduced (although still high compared to GRDC estimates of runoff) in the successive version 3.3 (Liman et al., 2018). In fact, the mean over-ocean HOAPS-3.3 E-P, determined between $70°$ S–$70°$ N and 1988–2012, is $0.45$ mm d$^{-1}$, similar to the value we compute for the same time and latitude range using HOAPS-4.0, of $0.51$ mm d$^{-1}$. For the time and spatial range in the current study, 1997–2013 and within $80°$ of latitude, HOAPS-4.0 mean $E - P = 0.49$ mm d$^{-1}$ ($62 \cdot 10^3$ km$^3$ yr$^{-1}$), about $50\%$ larger than the GRDC estimate of $41 \cdot 10^3$ km$^3$ yr$^{-1}$. Nevertheless, the over-ocean freshwater fluxes of all studied data sets agree with each other and with runoff data within the HOAPS, SEAFLUX, and OAFlux uncertainty ranges.

The differences in over-ocean mean $P$ between ERA5, GPCP, and HOAPS can be traced back to differences in their probability density functions. HOAPS has a smaller probability for yielding intermediate rain rates, whereas GPCP yields less occurrences of large rain rates (Masunaga et al., 2019). In addition, HOAPS shows a much higher fraction of monthly, $1°$x$1°$ non-raining grid boxes (3-4%) than either GPCP (0.5-1%) or ERA5 (0.2%), which has a large impact on the mean value of $P$. The inter-comparison of global P data sets is the topic of a range of papers, but since the validation of $P$ is difficult due to its inherent variability and the lack of sufficient *in situ* data — particularly over ocean — judging which algorithm performs best under which circumstances is a complicated task (Kidd and Huffman, 2011; Gehne et al., 2016; Tapiador et al., 2017).

Since the studied data sets contain values over ocean only, it is not possible to check if total $E$ and $P$ balance globally. For this reason we include ERA5 reanalysis data into the comparison. Model physics parameterizations and dynamics presumably act to ensure that the large positive biases found in both ERA5 $E$ and $P$ (compared to satellite data) cancel out almost completely and ERA5 $E - P$ is in good agreement with most satellite data at latitudes $\leq 45°$. We show that ERA5's water budget is closed for the studied time range (1997–2013) and that the various components — E, P, TCWV tendency — are consistent on a monthly, global scale (Fig. 7). Global total VIMD, however, does not equal zero, which is due to the numerical method used to compute VIMD. For studies of the global water cycle using ERA5 data, we recommend the use of VIWVD instead, as its global total is closer to zero and its totals over land and ocean are in better agreement with each other and with results from our and previous studies (Table 4). Cautiously interpreting this consistency as an indication of good quality, we use ERA5 data to devise methods to examine the consistency of ocean-only satellite E and P data sets. The high correlation coefficient found for the regression of ERA5 $E_{ocean} - \nabla \cdot (vq)_{ocean}$ with ERA5 $P_{ocean}$ implies a high degree of coherence, yet correlations of satellite E data with GPCP or HOAPS P are small (Table 3). This is certainly partly due to the number of different sources

of data, which for ERA5 is 1, but for (e.g.) J-OFURO is 3: J-OFURO E, ERA5 VIMD, and GPCP P, each having its own sampling characteristics and uncertainties. But the lack of correlation is probably also caused in part by an actual lack of coherence between satellite E data and GPCP (or HOAPS) P. This, in turn, implies that inaccuracies in satellite E and/or P data remain that may prevent closure of the over-ocean part of the water cycle. The comparison of estimates of total $E_{ocean}$ and $P_{ocean}$ with estimates of transport, continental runoff, and $(E - P)_{land}$ (Table 4) paints a similar picture: over-ocean E and P show a large spread in values, coupled with high uncertainties.

## 6 Final Comments

Our inter-comparison of six CDRs shows agreement among global means of $E - P$ within HOAPS-4.0, OAFlux3, and SEAFLUX3 uncertainty ranges. Despite considerable positive biases in ERA5 $E$ and $P$, over-ocean ERA5 $E - P$ is in agreement with satellite data, showing some temporal coherence in variations on monthly–decadal time scales, but with notable departures depending on time and on the E data set used. Within uncertainty, over-ocean total $E - P$ from satellites is in agreement with estimates of continental runoff and net ocean-to-land transport. However, uncertainties of and the spread among satellite data sets are both still very large in comparison with the magnitude of over-ocean $E - P$. Improving estimates of $E$ and $P$, particularly over ocean, thus remains an important task. Moreover, emphasis should be put on the development of uncertainty ranges. We recommend that to monitor the quality of results, in addition to performing independent validation studies, the whole global water cycle and the constraints it imposes should be taken into consideration.

The presented framework is based on co-variation of water cycle components and global water budget constraints. We applied it to the inter-comparison of satellite observations, but it can also be used for climate model assessments such as CMIP (see, e.g. Held and Soden, 2006; Liepert and Previdi, 2012; Knutti and Sedláček, 2013; Allan et al., 2020).

There is a pressing need to understand the nature of changes to the Earth's water cycle induced by global warming. The consensus in recent scientific literature is that there will be a larger amount of water vapor in the atmosphere as the atmosphere warms and, consequently, its water-holding capacity increases at a rate consistent with the Clausius-Clapeyron relationship (e.g, Allen and Ingram, 2002; Held and Soden, 2006; Shie et al., 2006; Trenberth et al., 2007; Allan et al., 2020). Model simulations agree on E and P flux responses to SST change of about 2%–3% K$^{-1}$ (Allan et al., 2020), but observational confirmation through satellite estimates is only now emerging from the background of noise from natural climate variability. We here show that ERA5, a state-of-the-art reanalysis, underestimates seasonal and inter-annual variability of E-P compared to satellite-based observations, which is also the case for climate models (Wentz et al., 2007). This could tentatively be interpreted as indicating that the water cycle is more sensitive to short-term changes in the state of the atmosphere and ocean than models predict. However, the stability of observations is affected by changes in satellite observing system. These changes, combined with assumptions contained in algorithms for near-surface humidity and wind speed (needed for bulk aerodynamic retrievals) complicate the detection and quantification of long-term trends (Wentz et al., 2007; Trenberth et al., 2007; Schlosser and Houser , 2007; Robertson et al., 2014). Moreover, and despite the fact that the satellite record of water cycle components now

encompasses more than three decades' worth of data, changes in E-P expected from (anthropogenic) global warming within this time period are weak compared to natural changes (Allen and Ingram, 2002; Allan et al., 2020).

In general, the quality of observations of the water cycle needs to improve before attempts at assessing effects of climate change from those data can be undertaken. The importance of accompanying high-quality uncertainty information cannot be overstated.

*Data availability.* All presented data sets are freely available from the cited websites.

## Appendix A: Difference climatologies

All difference maps of satellite-based E-P, E, and P climatologies with collocated ERA5 data are shown in Fig. A1. The maps in the left column are very similar (apart from HOAPS) because the E-P deviations are dominated by P and the same GPCP data were used to generate all E-P data sets (except HOAPS).

## Appendix B: Regression of ERA5 E-P against $\nabla \cdot (vq)$

Locally and over long time scales (e.g. one month) E-P and VIMD are equal (Eq. 3), as the change in TCWV is negligible on those scales. To see if this is the case for ERA5, the upper panels of Fig. A2 show the correlation coefficient, slope, and intercept of the linear regression of monthly mean E-P with VIMD. A linear fit given by $\nabla \cdot (vq) = a(E-P)+b$ yields a slope $a$ very near to $1.0$ and an intercept $b$ close to $0$ mm d$^{-1}$ everywhere, with a tendency to $a > 1$ over high $P$ regions (e.g. the ITCZ) and $a < 1$ elsewhere, as shown in the middle and right panels of Fig. A2. Due to errors introduced during data processing (e.g., by moving between spectral and grid-point space) a perfect match between the ERA5 variables is not expected. But there are a few regions where the decreased $R^2$ can only be partly ascribed to averaging errors. These are the oceans' desert regions, e.g., at the Peruvian coast and southern Africa's west coast, where climatological mean $P$ is less than $1$ mm d$^{-1}$, and the dynamic range of $E - P$ is small ($\leq 2$ mm d$^{-1}$). In the ocean deserts, the slope is $< 1$ and the intercept $> 0$ mm d$^{-1}$ (e.g., $0.75$ and $1.5$ mm d$^{-1}$ for the tropical East Pacific region with small $R^2$). Since mean $P$ is near $0$ mm d$^{-1}$ in these regions, $\nabla \cdot (vq)$ is approximately equal to $E$. Hence the deviations between $E-P$ and $\nabla \cdot (vq)$ in these regions indicate an inconsistency between ERA5 $E$ and $\nabla \cdot (vq)$. From the similarity of ERA5 $E-P$ and $\nabla \cdot (vq)$ it follows that the results of the regression analysis of ERA5 E-P with SEM E-P, presented in Sect. 4.3, are very similar to those obtained for ERA5 $\nabla \cdot (vq)$ with SEM $E-P$, as shown in the lower panels of Fig. A2. The correlation coefficient is somewhat smaller than for ERA5 $E-P$, but the patterns of all three statistical parameters are very similar to those in the last row of Fig. 6.

*Author contributions.* M.S. initiated the study on E-P intercomparisons and M.G. on water cycle closure; M.G. designed and performed the analyses; M.G., M.S., K.F., T.T., S.B., J.B.R., and F.R.R. discussed results; M.G. wrote the paper with contributions from all.

*Competing interests.* There are no competing interests.

*Acknowledgements.* We gratefully acknowledge Michael Mayer and two additional anonymous referees, whose numerous and detailed comments and recommendations helped improve the manuscript. We are also grateful to Hans Hersbach, Paul Berrisford, and Anton Beljaars
(ECMWF) for their helpful comments on ERA5 assimilation and parameterization. Hannes Konrad (DWD) is acknowledged for stimulating discussions and Frits Penning de Vries for critical reading of the manuscript.

K.F. and M.S. acknowledge financial support by the EUMETSAT member states through CM SAF.

Version 1.3 GPCP-1DD combined precipitation data were provided by the NCEI CDR Program as a contribution to the GEWEX Global Precipitation Climatology Project. Global ocean heat flux and evaporation products were provided by the Woods Hole Oceanographic Insti-
tution (WHOI) OAFlux project funded by the NOAA Climate Observations and Monitoring program. We thank IFREMER, the J-OFURO project and the SeaFlux project at WHOI for sharing their data free of charge. ERA5 results shown here were generated using data from Copernicus Climate Change Service Information (2020).

Finally, we gratefully acknowledge developers and contributors to Python and Stackexchange.

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

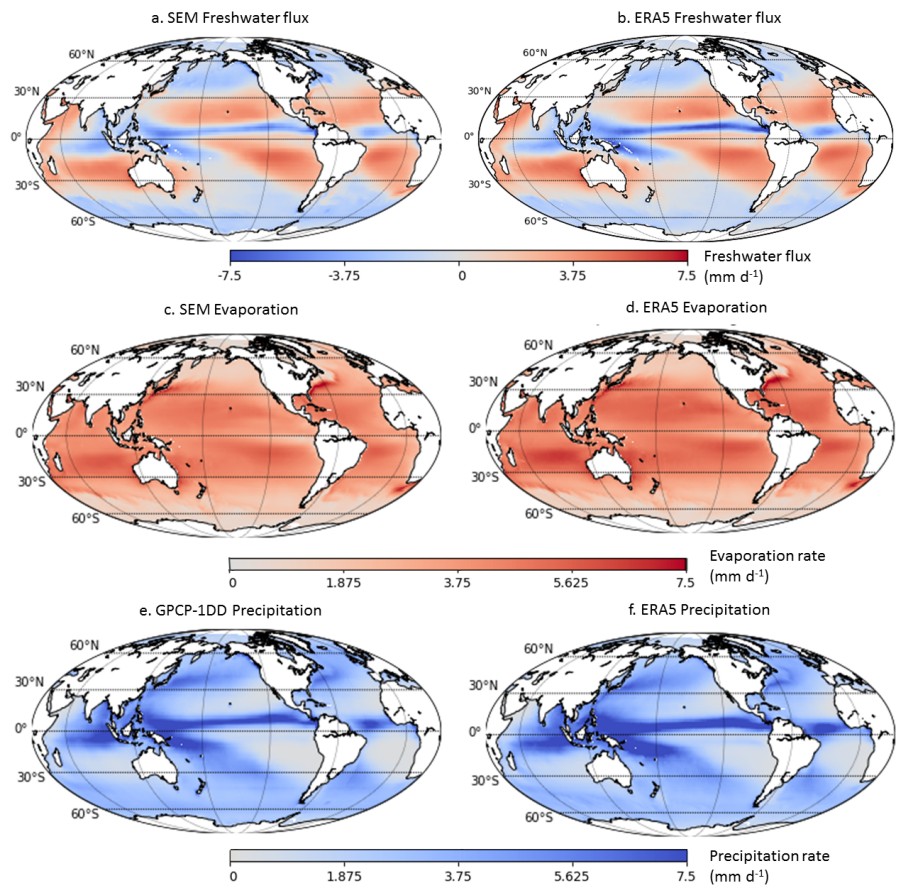

**Figure 1.** Satellite ensemble median (SEM) and ERA5 climatologies (1997-2013) of freshwater flux (a and b) and evaporation (c and d), and GPCP and ERA5 precipitation (panels e and f). ERA5 data coverage was reduced to match satellite data, and data over land were discarded from panels e and f. See the text for details.

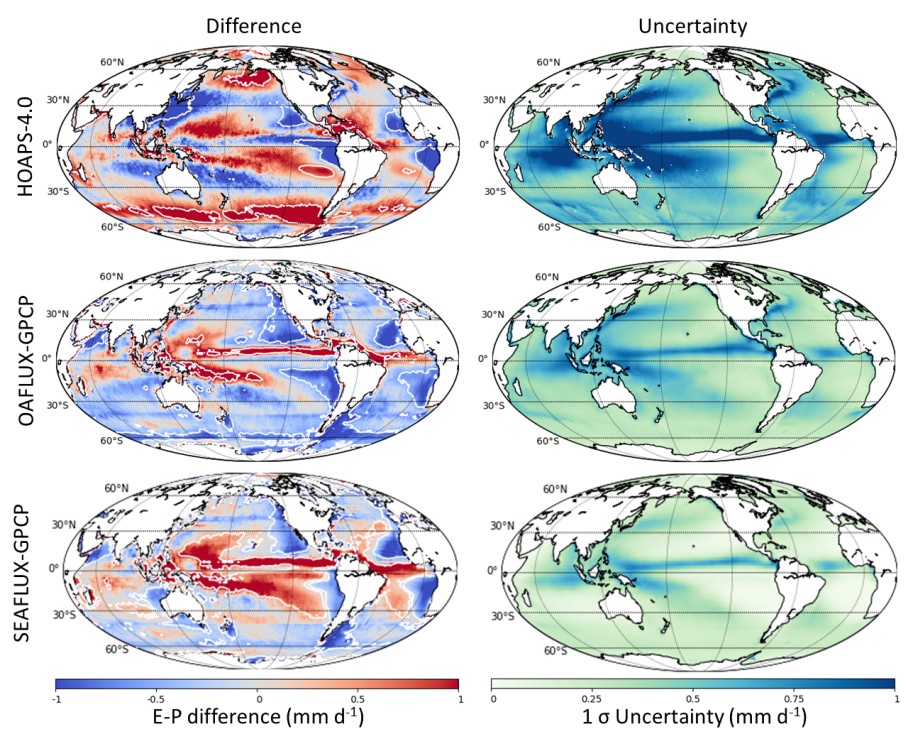

**Figure 2.** Left panels: Difference maps of HOAPS (upper), OAFlux-G (center), and SEAFLUX-G (lower) climatological mean E-P minus the corresponding collocated ERA5 climatology (1997-2013). Right panels: HOAPS (upper), OAFlux-G (center), and SEAFLUX-G (lower) climatological mean $1\sigma$ uncertainty. White lines in the left panels enclose regions where the difference with ERA5 E-P exceeds the $2\sigma$ uncertainty range.

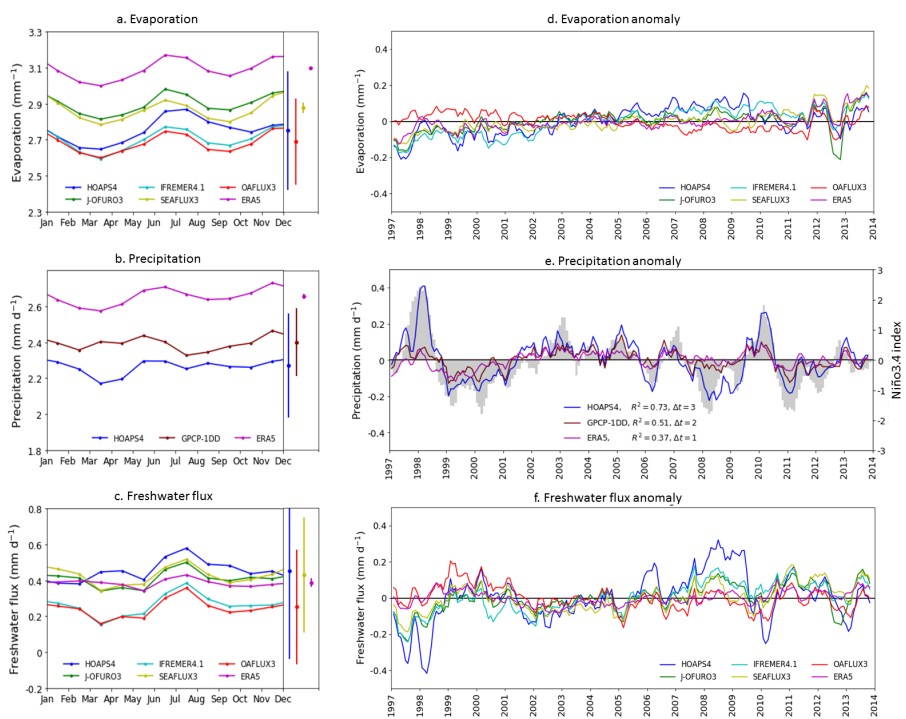

**Figure 3.** Climatological (1997–2013) seasonal cycle of global ocean mean evaporation rate (a), precipitation rate (b), and freshwater flux (c). HOAPS, ERA5, OAFlux, SEAFLUX, and GPCP mean values and associated $1\sigma$ uncertainty ranges are shown in the boxes to the right of the panels. Monthly mean anomaly (w.r.t. the climatological seasonal cycle depicted at left) over the global oceans (80° S–80° N) of evaporation rate (d), precipitation rate (e), and freshwater flux (f). The anomaly data are smoothed using a three-month running mean. Panel e additionally displays the Niño3.4 index shifted by +3 months (right y-axis). The legend shows the correlation coefficient of the Niño3.4 index with P anomalies and the time lag of highest correlation ($\Delta t$ in months). Ticks on the time axis mark January of the indicated year.

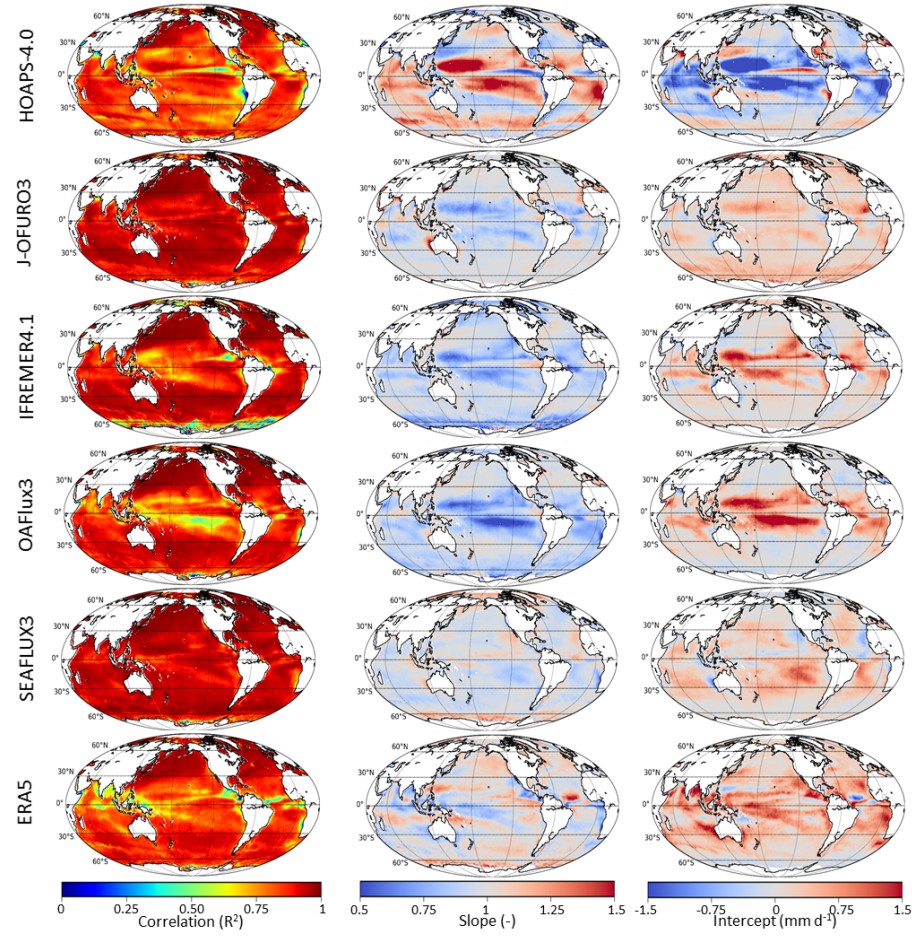

**Figure 4.** Correlation, slope and intercept of the linear regression of monthly mean $E$ from (top to bottom): HOAPS, J-OFURO, IFREMER, OAFlux, SEAFLUX, or ERA5 with satellite ensemble median (SEM) monthly mean $E$ (1997–2013).

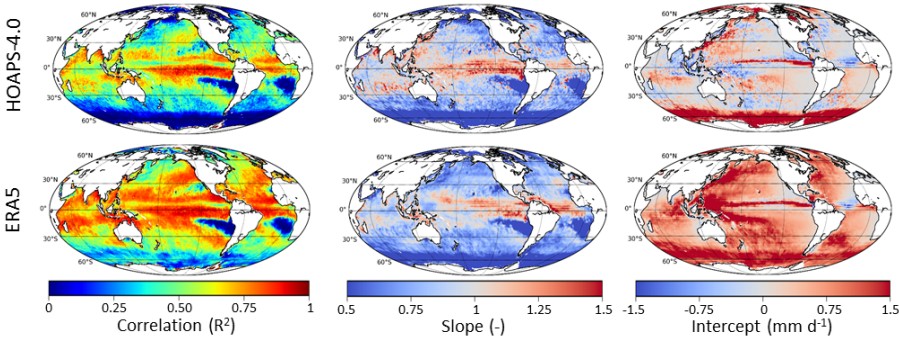

**Figure 5.** Correlation, slope and intercept of the linear regression of monthly mean $P$ from HOAPS (upper panels) or ERA5 (lower panels) with GPCP monthly mean $P$ (1997–2013).

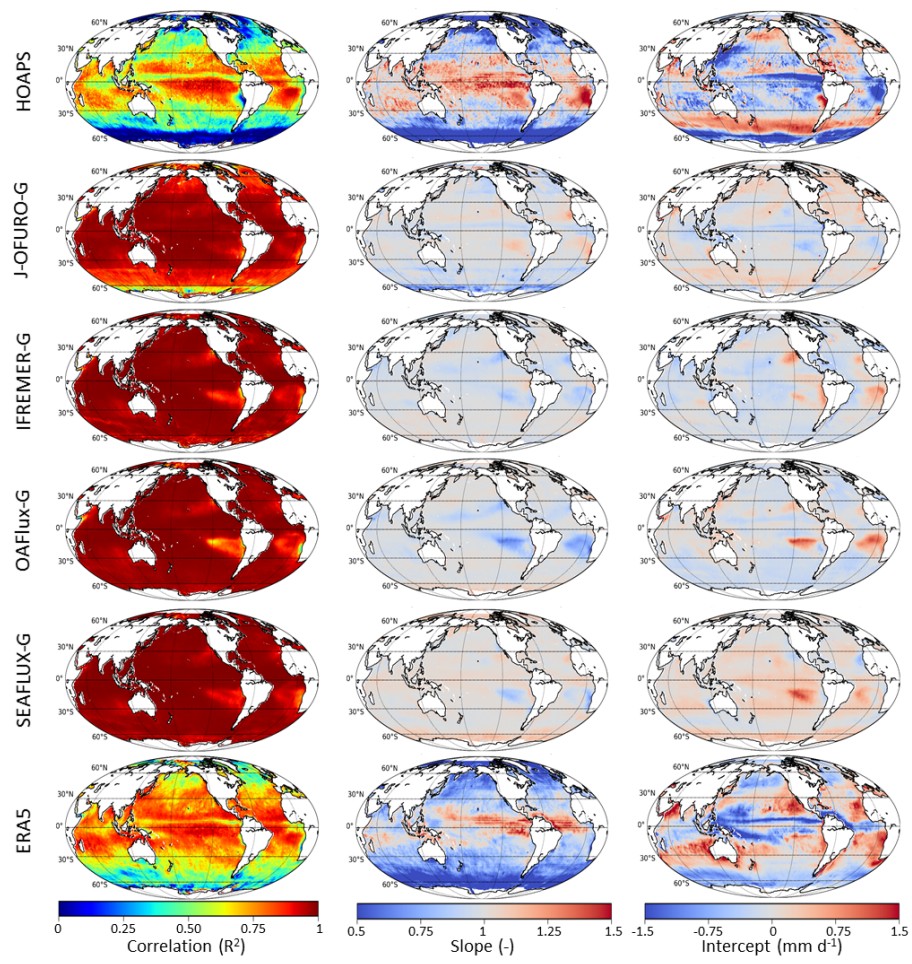

**Figure 6.** Correlation, slope and intercept of the linear regression of monthly mean $E - P$ from (top to bottom): HOAPS, J-OFURO-G, IFREMER-G, OAFlux-G, SEAFLUX-G, or ERA5 with satellite ensemble median (SEM) monthly mean $E - P$ (1997–2013).

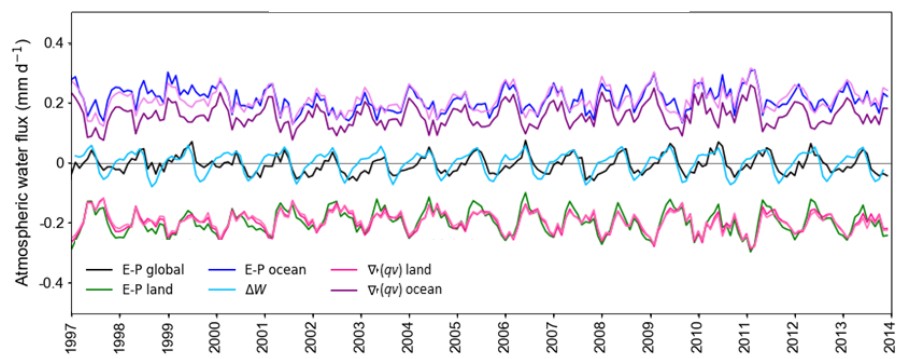

**Figure 7.** ERA5 monthly mean $E - P$ over the whole globe (black), land only (green), and ocean (blue); global mean $\Delta W$ (light blue), mean $\nabla \cdot (vq)$ over land (pink) and ocean (purple). The mean values over the globe and land were scaled by their surface area relative to the ocean surface area (i.e., they were multiplied by $510/350$ and $160/350$, respectively) to obtain consistency with the over-ocean means shown in Fig. 3. Error bars represent the standard deviation within the 10-member ensemble, which is smaller than the graph's line width for $E - P$ over land, $\Delta W$, and $\nabla \cdot (vq)$.

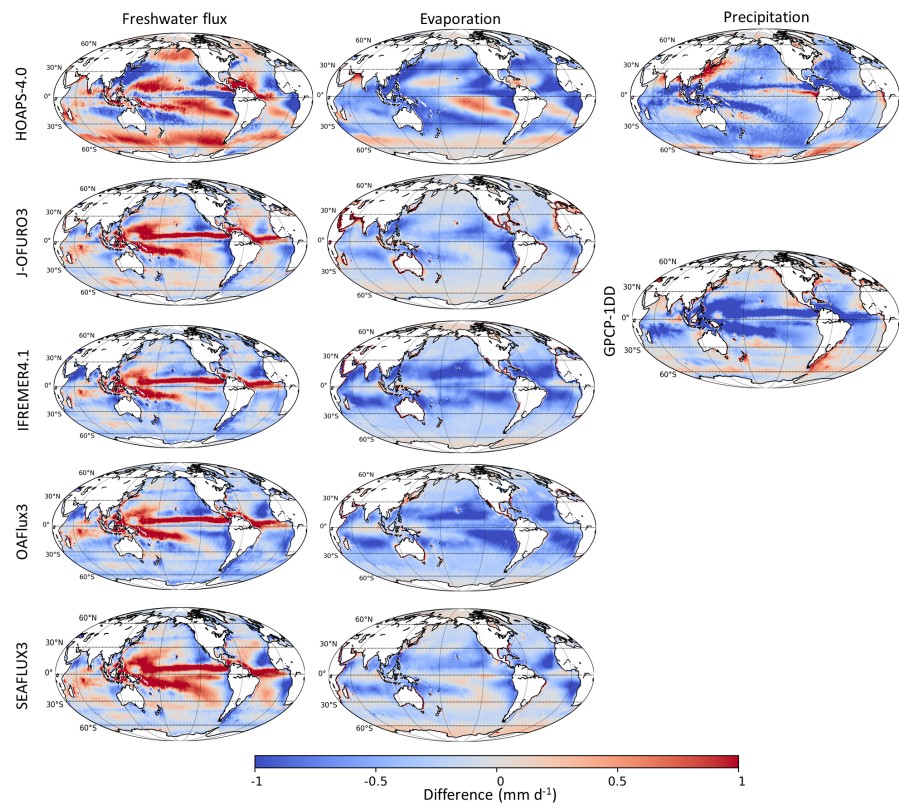

**Figure A1.** Difference maps of satellite-based E-P (left), E (center), and P (right) climatologies and the respective ERA5 climatology (1997-2013).

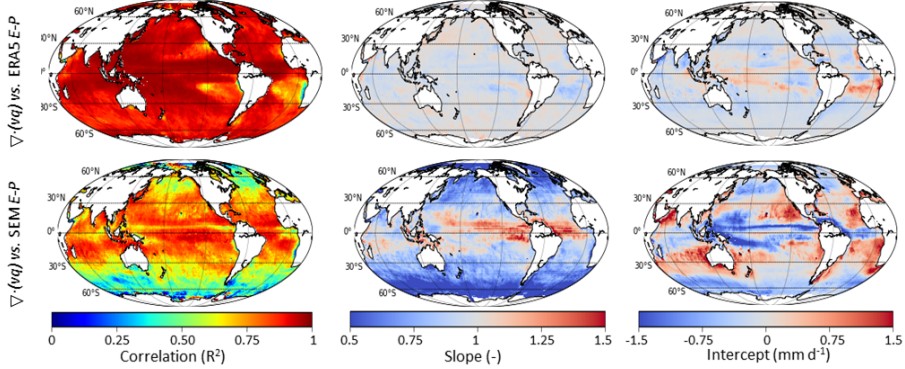

**Figure A2.** Correlation, slope and intercept of the linear regression of monthly mean ERA5 $\nabla \cdot (vq)$ with ERA5 $E - P$ (upper panels) or with the satellite ensemble median (SEM) $E - P$ (lower panels).

**Table 1.** Compilation of the data sets used within this study. Most data sets contain more variables than those listed here.

| Acronym | Dataset Name | Date Range | Resolution | Variables | Reference |
|---------|--------------|------------|------------|-----------|-----------|
| ERA5 | ECMWF Reanalysis 5 | 01/1979 to present | 0.25°x0.25° | E, P, TCWV, VIMD | Hersbach et al. (2020) |
| GPCP-1DD V 1.3 | Global Precipitation Climate Program | 01/1996 – 12/2017 | 1.0°x1.0° | P | Huffman et al. (2001) |
| HOAPS V4.0 | Hamburg Ocean Atmosphere Parameters and Fluxes from Satellite | 07/1987 – 12/2014 | 0.5°x0.5° | E, P, E-P | Andersson et al. (2010) |
| J-OFURO V3 | Japanese Ocean Flux Data Sets with Use of Remote-Sensing Observations | 01/1988 – 12/2013 | 0.25°x0.25° | LHF, E-P | Tomita et al. (2018) |
| OAFlux V3 | Objectively Analyzed Air-sea Fluxes[1] | 1958–2019 (monthly) 1985–2017 (daily) | 1.0°x1.0° | E | Yu et al. (2008) |
| IFREMER V4.1 | Institut Français de recherche pour l'exploitation de la mer | 01/1992 – 12/2018 | 0.25°x0.25° | LHF, SST | Bentamy et al. (2013) |
| SEAFLUX V3 | Sea Flux Project | 01/1988 – 12/2018 | 0.25°x0.25° | LHF, SST | Roberts et al. (2020) |

[1] https://www.esrl.noaa.gov/psd/data/gridded/data.oaflux_v3.html

**Table 2.** Abbreviations and symbols of variables used throughout the manuscript.

| Variable | Abbreviation | Symbol |
|---|---|---|
| Air density | - | $\rho$ |
| Evaporation rate | E | $E$ |
| Latent heat flux | LHF | $Q_l$ |
| Near-surface (10 m) humidity | - | $q_a$ |
| Near-surface (10 m) wind speed | - | $u$ |
| Precipitation rate | P | $P$ |
| Runoff | R | $R$ |
| Sea surface humidity | - | $q_s$ |
| Sea surface temperature | SST | $T_s$ |
| Latent heat of evaporation of water | - | $L_E$ |
| Total column water vapor | TCWV | $W$ |
| TCWV tendency | $\Delta$TCWV | $\Delta W$ |
| Turbulent exchange coefficient | - | $C_E$ |
| Vertically integrated moisture flux divergence | VIMD | $\nabla \cdot (vq)$ |

**Table 3.** Pearson's correlation coefficient squared ($R^2$) for monthly (mean or anomaly) or yearly global ocean mean $E_{ocean} - \nabla \cdot (vq)$ *vs.* $P_{ocean}$, with $\nabla \cdot (vq)$ data from ERA5. $R^2$ was calculated from data sets that were collocated prior to the calculation of global means. Non-significant correlation coefficients (p-value $> 0.05$) are marked with an asterisk.

| Data set | monthly mean | yearly mean | monthly anomaly |
|---|---|---|---|
| HOAPS-4.0 | 0.03 | 0.00* | 0.06 |
| J-OFURO3 - GPCP-1DD | 0.16 | 0.31 | 0.22 |
| IFREMER4.1 - GPCP-1DD | 0.13 | 0.23 | 0.20 |
| OAFlux3 - GPCP-1DD | 0.14 | 0.01* | 0.11 |
| SEAFLUX3 - GPCP-1DD | 0.17 | 0.02* | 0.12 |
| ERA5 | 0.86 | 0.86 | 0.83 |

**Table 4.** Estimates of ocean total $E$ and $P$, land and ocean total $E - P$, net transport of water vapor, and continental runoff given in $10^3$ km$^3$ yr$^{-1}$. The upper three rows contain results from this study, the lower five those from earlier investigations. ERA5 estimates are calculated from ensemble mean data, the standard deviation (std) is derived from ensemble statistics. The satellite-based data sets used in our study were averaged to obtain the mean and std of observed (Obs.) $E_{ocean}$ and $P_{ocean}$, and the range is given in the third row. Net water vapor flux divergence over land ($\nabla \cdot (vq)_{land}$) and ocean ($\nabla \cdot (vq)_{ocean}$) and continental runoff $R$ are given in the last three columns. The estimates from the study by Rodell et al. (2015) are separated into observations (obs.) and model-optimized observations (opt.), see the text for details.

| | $E_{ocean}$ | $P_{ocean}$ | $(E-P)_{ocean}$ | $(E-P)_{land}$ | $\nabla \cdot (vq)_{land}$ | $\nabla \cdot (vq)_{ocean}$ | $R$ |
|---|---|---|---|---|---|---|---|
| ERA5 | $467 \pm 1$ | $426 \pm 2$ | $43 \pm 2$ | $-44 \pm 0.4$ | $-43 \pm 0.2$ | $31 \pm 0.2$ | 42.1 |
| Obs. mean $\pm$ std | $425 \pm 20$ | $360 \pm 25$ | $52 \pm 13$ | – | – | – | – |
| Obs. range | 397–453 | 335–384 | 35–65 | – | – | – | – |
| Oki and Kanae (2006) | 436.5 | 391 | 45.5 | $-45.5$ | $-45.5$ | 45.5 | 45.5 |
| Trenberth and Asrar (2014) | 413 | 373 | 40 | $-40$ | $-40$ | 40 | 40 |
| Rodell et al. (2015) obs. | $410 \pm 36$ | $385 \pm 39$ | $24 \pm 53$ | $-45 \pm 10$ | $-43 \pm 8$ | $47 \pm 19$ | $50 \pm 7$ |
| Rodell et al. (2015) opt. | $450 \pm 22$ | $403 \pm 22$ | $46 \pm 31$ | $-46 \pm 7$ | $-46 \pm 4$ | $46 \pm 2$ | $46 \pm 4$ |
| Allan et al. (2020) | $480 \pm 48$ | $434 \pm 43$ | $46 \pm 65$ | $-46 \pm 14$ | $-46 \pm 5$ | $46 \pm 5$ | $51 \pm 3$ |