# Peer review of "Intercomparison of freshwater fluxes over ocean and investigations into water budget closure"

_Hydrology and Earth System Sciences, 2020_

## Referee Comment (RC1) · Anonymous Referee #1 · 28 Aug 2020

Review of: Intercomparison of freshwater fluxes over ocean and investigations into water budget closure By Gutenstein et al. This paper presents an inter-comparison of five recent satellite-based and one re-analysis E-P data sets. The different data-sets and the assumptions behind them are described. The different components of the hydrological cycle are presented separately. This is a well written paper, which presents a valuable contribution to the climate community. I have little to add to this paper, which in my opinion is almost ready for publication in its current form. The few and very minor comments I have are: • In the introduction I missed a section motivating the study from a climate change perspective like you added to the "Final Comments" section. Monitoring trends in the hydrological cycle is of great importance under uncertain

changing climate conditions. In that aspect I would like to point the authors to some recent papers on the topic 1-3 • The inter-comparison presented here is a very nice and useful framework also for comparing with climate models 4-6 such as CMIP6. I suggest to propose it in the "Final Comments" section (or elsewhere) for future work. It could enlarge the connection of this work to climate change research. • L96: Could you please elaborate on how wind speed is calculated based on temperature (BT) measurements? In addition, If E estimates are based on BT measurements, which are more accurate in clear sky conditions than in cloudy sky conditions, wouldn't that cause a bias? How is it calculated in cloudy (and rainy) conditions? If it is only calculated in clear sky conditions, wouldn't the E estimations be biased high (as in cloudy and rainy conditions the evaporation is lower)? • L368: is the largest deviation in E estimations in the tropics due to the large (and optically thick) cloud cover? • L437: I think that the correlation does not decrease when Delat Qocean is not considered because there is basically no correlation even when it is considered. So, it can't get any lower than that. Is that correct? Technical comments: • L208: us–>use. • You alter between italic and non-italic in P, E and E-P. I think it should all be italic.

References 1 Allan, R. P. et al. Advances in understanding large-scale responses of the water cycle to climate change. Annals of the New York Academy of Sciences (2020). 2 Dagan, G., Stier, P. & Watson‐Parris, D. Analysis of the atmospheric water budget for elucidating the spatial scale of precipitation changes under climate change. Geophysical Research Letters (2019). 3 Yin, J. & Porporato, A. Looking up or looking down? Hydrologic and atmospheric perspectives on precipitation and evaporation variability. Geophysical Research Letters 46, 11968-11971 (2019). 4 Liepert, B. G. & Previdi, M. Inter-model variability and biases of the global water cycle in CMIP3 coupled climate models. Environmental Research Letters 7, 014006 (2012). 5 Knutti, R. & Sedláček, J. Robustness and uncertainties in the new CMIP5 climate model projections. Nature Climate Change 3, 369 (2013). 6 Held, I. M. & Soden, B. J. Robust responses of the hydrological cycle to global warming. Journal of Climate 19, 5686-5699 (2006).

---

## Referee Comment (RC2) · Anonymous Referee #2 · 12 Sep 2020

Review of "Intercomparison of freshwater fluxes over ocean and investigations into water budget closure" by M. Gutenstein et al.

This is a nice intercomparison study of various satellite-based precipitation and evaporation products and ERA5. The authors show that there is a large spread among the different products, and most of them fail to satisfy global budget constraints, especially when combining different products for estimation of P, E, and the transports. ERA5 performs best with a remarkably good agreement of forecast-based fluxes (P and E) and analysis-based transports (moisture flux divergence). The paper is well-structured and –written and therefore easy to follow. My only major comment is on the obvious

error in the computation of ERA5 moisture transports as detailed below. Moreover, the reader would probably like to see stronger conclusions. I know it is hard to make a ranking, but e.g. SEAFLUX with its clearly unphysical P-E over ocean could be ruled out as clearly unrealistic. Also HOAPS appears to be a bit of an outlier, especially in terms of variability. The budget constraints are an objective measure to rule out poor data, and this helps to better constrain the best estimate of the water budget, without unnecessary inflation of the error bars. Otherwise I only have a number of minor comments.

Major comment: The authors use VIMD from ERA5 to compute ocean-to-land moisture transports. For this it should not matter whether on integrates VIMD over all land points or over all ocean points. Another constraint is that the global average of VIMD must be zero. This is a mathematical constraint independent of data quality. However, the authors obtain inconsistent results for land and ocean integrals of VIMD (table 4). So either the archived VIMD fields are flawed (which I doubt as I am using the ERA5 data myself and could not find a similar problem) or there is some error in the author's processing chain that leads to these erroneous results. If this problem is really due to interpolation errors, as suspected by the authors, these interpolation errors are clearly unacceptably large. In short, this error must be corrected.

Minor comments: Equation 1 and everywhere else: as it stands, the VIMD terms looks like the moisture gradient. I suggest to replace with the more appropriate nabla * (vQ). L17: I presume you use monthly values. Please say it clearly, as the correlation strongly depends on the considered timescales. L37: The term "model reanalysis" seems an uncommon term to me. I suggest to drop "model". If you want to give an attribute, it may be better to say "climate reanalysis" or "dynamical reanalysis". L46: Isn't there an author on the GPCP document? L58: "moisture divergence" is sloppy terminology. Moisture itself cannot diverge. It should be "moisture flux divergence". L58: another nitpicky comment: VIMD is technically not identical with advection, although it is an excellent approximation of it. I suggest a slightly more cautious wording. L85-86:

[Figure]

"model runs" sounds it does not use any observational info. Simply say it consists of ten ensemble members. L108: Le is usually called "latent heat of evaporation". L133: Do you mean SST averaged over the top 0.5m? Please clarify. L183-184: The statement about forecast skill is hard to understand for a non-expert. L184: "model runs" → see comment above. L184: It should be mentioned that the ERA5 ensemble members have a lower resolution than the stated 30km. How would the results change when using the high-resolution ERA5 data? L189: I suggest to delete "on single levels" L192: It should be noted that monthly P and E from ERA5 is averaged from short-term forecasts (12 or 24 hours? Needs to be clarified as well!) L198: If the TCWV tendency is computed from monthly means (rather than instantaneous values at beginning and end of the month), one should use centered differences. L243: When deriving E from monthly Q fields, what is the error from neglect of sub-monthly covariance between E and SST? L258: I suggest to replace "relative" with "area-specific" and "total" with "area-integrated". L260: Please clarify how sea ice is treated in general. I presume it is masked out? Is this a seasonally varying mask? L273: "at the ITCZ" maybe better "in the ITCZ"? L280: Figure 2: One could make the simple statement that HOAPS differences are generally larger (RMS values of the field would be useful), but areas of disagreement are smaller because of the larger uncertainties. L285: similarly → similar L317: In terms of flow of reading, it may be better to move the sentence about ENSO correlation further down to around L340. Figure 3 and in general: I suggest to change the panel labelling to small letters, as capital letters have potential for confusion, especially "E". Best would be E→ (e) Figure 3: It would be interesting to see the ENSO correlation for every curve. This could be given in the legend, ideally with the lag at which the maximum correlation occurs. L319: Is "bias" the right term? We see differences, but still one of the datasets could be unbiased. L337: "biased low". SEAFLUX seems to be low in general (according to the mean annual cycle figure). So better to change to something like "particularly low". L373: Is there a reference for the statement on detection of snow in HOAPS? L389: remove "are" L413: This statement would be correct if VIMD was the 3D-divergence, i.e. including fluxes at top of the atmosphere, where there theoretically

could be an exchange with space. However, your VIMD is 2D and its global average is 0 according to the sentence of Gauss. L415: See my major comment. L503: I think it should be "right-most" L575-576: Please provide a reference for this statement. L586: This statement is on "observation-based attempts". Please clarify. Figure 4: middle and right columns: Would it be possible to use color schemes that are really white in the middle? Table 4: How is runoff from ERA5 obtained? Is this the area-integral of all grid point values?

---

## Referee Comment (RC3) · Anonymous Referee #3 · 14 Sep 2020

**Reviewer's Comments and Suggestions on Manuscript Entitled**

**"Intercomparison of freshwater fluxes over ocean and investigations into water budget closure"**

by Marloes Gutenstein, Karsten Fennig, Marc Schröder, Tim Trent, Stephan Bakan, J. Brent Roberts, and Franklin R. Robertson

**General Comments:**

It is an overall well-written manuscript comprehensively addressing a truly important subject on freshwater fluxes over ocean and (the associated) water budget closure by using and intercomparing seven credible and lengthy globally-covered products of evaporation and/or precipitation (six generally satellite-based datasets, along with one reanalysis). This reviewer would like to acknowledge the authors for their willing to challenge themselves and tackle such a subject that would almost (if not absolutely) guarantee and require tremendous efforts and time, and persistent commitments, let alone equipping with solid expertise and knowledge, and great ideas and insights.

However, a few relatively major concerns may need further elaboration or revisions. Several minor revisions are also suggested.

Finally, this reviewer considers that the authors deserve a solid credit, again for their tremendous efforts and crucial works. This reviewer will highly recommend this manuscript for publication by Hydrology and Earth System Sciences (HESS) (an Open Access Journal) should the following suggested revisions be properly conducted accordingly.

**Major Comments/Suggestions/Revisions:**

**1)** E-P uncertainty involving E uncertainty and P uncertainty:
The uncertainties of E-P of different sets of products involved and targeted in this study should mainly be depending on the respective uncertainties of the P and E products, e.g., for IFREMER-G(E-P) = IFREMER(E) - GPCP(P), the uncertainty of IFREMER-G(E-P) should supposedly be an added sum of the uncertainty of IFREMER(E) and the uncertainty of GPCP(P). These respective E and P uncertainties, however, are not among the main focuses of this study (as the authors have indicated). It'd be very understanding and foreseeable that such (additional) tasks of investigating and analyzing (or may even need to newly generate or estimate) the respective P and E uncertainties would add up another level of difficulty and efforts, especially if/when researchers were not directly involved in those P and E productions, and the currently available related uncertainty info's have been quite limited (which have also been revealed in this manuscript). As for "*We conclude that for a better understanding of the global water budget, the quality of E and P data sets themselves and their associated uncertainties need to be further investigated*", this reviewer has fully agreed on this critical "conclusion" finding, which, honestly speaking, has also been "expected" during the midst of review. It might also be fair and reasonable to alternately say "*for a better understanding of the global fresh water (E-P or P-E)*"

*distributions and their uncertainties, the quality of E and P data sets themselves and their associated uncertainties need to be further investigated*". As mentioned in the general comments, this reviewer has highly acknowledged the authors' tremendous efforts and time invested in this truly important and genuinely challenging "project" (must have involved multiple tasks!), it would be irrational if this reviewer would suggest the authors to further consider performing similar rounds of uncertainties analyses on P and E, respectively. However, this reviewer would like to suggest that the authors may consider drafting one additional paragraph or a set of sentences (and add it in, e.g., near the end of the Introduction section) including brief justified reasons (why the P and E uncertainties are not in the main focuses of this study) and useful messages (P and E, respectively, are by all means critical for the E-P quality and uncertainty). Of course, the authors' current concluding remarks may remain intact unless the authors may also wish to elaborate a bit more accordingly.

**2)** E-P uncertainty involving ∇Q and ΔW (a comment triggered by seeing the assumptions made in **Eqs. 1-3**)

The assumptions of neglecting ∇Q or ΔW for global or regional scales have trigged this reviewer wondering about their potential impact on the E-P uncertainty. Here's the comment that the authors may feel free to respond or not (**Optional**).
In the case of omitting ∇Q or ΔW, it might cause two folds of potential impacts, hypothetically: 1) if neither ∇Q nor ΔW would carry uncertainties, then the currently estimated uncertainty of E-P in this study could have been overestimated since part of the estimated uncertainty might have been implicitly contributed by the being-omitted "true" amounts of ∇Q or ΔW (even though being small, but had been neglected), 2) if either/both ∇Q or/and ΔW would carry uncertainties, then the currently estimated uncertainty of E-P in this study could have been underestimated, especially if/since the uncertainties of ∇Q or ΔW (if available and "certain") would be sufficiently large (greater than their true amounts) and add an additional net amount of uncertainty into E-P, otherwise. The E-P uncertainty could have still been overestimated as in "1)" if the true amounts of ∇Q or ΔW are larger than their respective uncertainties.

**3) On line 272** (related to **Fig. 1**):
A seemingly correct but actually questionable (not quite proper) statement that needs a "major" attention and revision.
"*A and B shows that the E-P pattern is mainly determined by P, as there is less spatial variation in E.*"
It should be more accurate to state "*A and B shows that the E-P pattern is mainly determined by P in the tropical and high-latitude regions, but determined by E in the subtropical regions.*" It's also the absolute magnitudes/intensities of E or P, not just only their spatial variation that would matter. Actually, if "warm" color (red) and "cold" color (blue) were applied for E (panels C and D) and P (panels E and F), respectively, such crucial features (reviewer's points) would have been better revealed. Perhaps, the authors may genuinely consider it.

**4)** The authors would sometimes describe, elaborate or discuss the features or findings shown via figures or panels, but without consistently and explicitly citing them along the presentation/writing.

Here are a set of examples related to a few **Fig. 3 panels**:

- **On line 310**: may add in "*In panel A*" to "*E data from HOAPS, IFREMER, and OAFlux are much closer to each other…*"
- **On line 325**: may add in "*In panel C*" to "*Apart from HOAPS E−P in March–April….*"
- **On line 337**: may add in "*In panel E*" to "*The three P data sets yield inter-annual variations with amplitudes…*"
- **On line 339**: may add in "*In Panel F*" to "*Apparent agreement is found among all E−P anomalies…*"

It's also thus suggested that authors may want to apply similar revisions elsewhere consistently. Such changes should help reader's reading and comprehension effectively.

**Minor Revisions:**

**1) On line 46**:

Suggest change
"*are the most widely used data sets.*"
→
"*are among the most widely used data sets.*"

**2) On lines 58-59**:

Suggest change
"*With W the total column water vapor and $\nabla Q$ the moisture divergence, i.e., the amount of moisture removed by-advection from the considered volume.*"
→
"*With W the total column water vapor and $\nabla Q$ the total moisture divergence or convergence, i.e., the amount of moisture associated with the moisture advection and the mass divergence or convergence scaled by water vapor from the considered volume.*"

**3) On line 157**:

Suggest change
"*Version 2 (GSSTF2.0, Shie et al. (2009))*"
→
"*Version 2; 2c (GSSTF2.0, Chou et al. (2003); GSSTF2c, Shie et al. (2009))*"

and the following respective reference should thus be included:

Chou, S.-H., E. Nelkin, J. Ardizzone, R. Atlas, and C.-L. Shie, 2003: Surface Turbulent Heat and Momentum Fluxes over Global Oceans Based on the Goddard Satellite Retrievals, Version 2 (GSSTF2), *J. Climate*, 16, 3256-3273. [10.1175/1520-0442(2003)016<3256:STHAMF>2.0.CO;2]

**4) On line 206**:

Suggest revise
"*see, e.g., Kidd and Huffman, 2011; Tapiador et al. , 2017)*"
→
"*see, e.g., Kidd and Huffman, 2011; Tapiador et al., 2017)*"

**5) On lines 207-208** :
Suggest revise
"*which makes us of its own P data)*"
→
"*which makes use of its own P data)*"

**6) On line 210**:
Suggest change
"*The Global Precipitation Climatology Project - 1 Degree Daily (GPCP-1DD; denoted GPCP hereafter)*"
→
"*The Global Precipitation Climatology Project - 1 Degree Daily (denoted GPCP and GPCP-1DD, respectively, hereafter)*"

**7) On line 235**:
Suggest revise
"*consisting of 10 seperate model runs*"
→
"*consisting of 10 separate model runs*"

**8) On line 389**:
Suggest revise
"*we are also examine the separate contributions*"
→
"*we also examine the separate contributions*"

**9) On line 458**:
Suggest revise
"*(Oki and Kanae , 2006 *"
→
"*(Oki and Kanae, 2006;*"

**10) On line 574**:
Suggest revise
"*(e.g, Allen and Ingram , 2002; Held and Soden , 2006; Trenberth et al., 2007).*"
→
"*(e.g., Allen and Ingram, 2002; Held and Soden, 2006; Trenberth et al., 2007).*"

**11) In Fig. 3B**:
There are four kinds of shading shown with three precipitation data sets.  The extra "dummy" shading should be removed.

**12) On line 573-574**:

"*its water-holding capacity increases at a rate consistent with the Clausius-Clapeyron relationship (e.g, Allen and Ingram , 2002; Held and Soden , 2006; Trenberth et al., 2007*)"

Here's one paper (i.e., Shie et al., 2006) that also addressed a Clausius-Clapeyron scenario focusing on tropical oceans. It may be considered and included as one reference.

Shie, C.-L., W.-K. Tao, and J. Simpson, 2006: A note on the relationship between temperature and water vapor over tropical oceans, including sea surface temperature effects, *Special Issue of Advances in Atmospheric Sciences*, Vol. 3, No. 1, 141-148. doi: 10.1007/s00376-006-0014-5

Electronic copy may be accessible at
http://www.iapjournals.ac.cn/fileDQKXJZ/journal/article/dqkxjz/2006/1/PDF/231scl.pdf

---

## Author Comment (AC1) · 21 Oct 2020

Authors' reply to Anonymous Referee #1

We thank the referee for his/her helpful comments to and kind words about our manuscript. We have incorporated most of the comments into the revised manuscript, as detailed below. Referee comments appear in italics, our reply in normal font, and changes to the manuscript in blue.

*Review of: Intercomparison of freshwater fluxes over ocean and investigations into water budget closure By Gutenstein et al. This paper presents an inter-comparison of five recent satellite-based and one re-analysis E-P data sets. The different data-sets and the assumptions behind them are described. The different components of the hydrological cycle are presented separately. This is a well written paper, which presents a valuable contribution to the climate community. I have little to add to this paper, which in my opinion is almost ready for publication in its current form.*
Thank you!

*The few and very minor comments I have are:*
*In the introduction I missed a section motivating the study from a climate change perspective like you added to the "Final Comments" section. Monitoring trends in the hydrological cycle is of great importance under uncertain changing climate conditions. In that aspect I would like to point the authors to some recent papers on the topic [1-3].*
The referee is right. We added the following paragraph to the introduction:
„At long temporal and/or large spatial scales, the increases in E and P with rising global temperature are relatively small (2-3%K$^{-1}$) and are constrained by the energy budget. At smaller scales (less than approximately 4000 km and/or 10 years) these changes can be much larger (or smaller) due to dynamical contributions (Dagan et al., 2019; Yin and Porporato, 2019; Allan et al., 2020). The nature and extent of these changes, which affect the livelihoods of many millions of people, are difficult to model due to various counteracting influences such as forcing by clouds and aerosols, or land use change (Allan et al., 2020). Close monitoring of E and P by (satellite) observations thus yields an important contribution to a better understanding of impacts of climate change at regional and local scales."

*The inter-comparison presented here is a very nice and useful framework also for comparing with climate models [4-6] such as CMIP6. I suggest to propose it in the "Final Comments" section (or elsewhere) for future work. It could enlarge the connection of this work to climate change research.*
That is also a good suggestion. We added the following statement to the Final Comments section:
„The presented framework is based on co-variation of water cycle components and global water budget constraints. We applied it to the inter-comparison of satellite observations, but it can also be used for climate model assessments such as the Coupled Model Intercomparison Project CMIP (see, e.g., Held and Soden, 2006; Liepert and Previdi, 2012; Knutti and Sedlacek, 2013; Allan et al., 2020)."

*L96: Could you please elaborate on how wind speed is calculated based on temperature (BT) measurements?*
Wind speed cannot be directly derived from passive satellite observations. However, wind effects alter the roughness and emissivity of the ocean surface, particularly affecting the 19 and 37 GHz channels (see, e.g., Meissner and Wentz, 2012). Just like heat fluxes (and evaporation rate), near-surface wind speed cannot be determined from passive microwave observations in scenes with heavy precipitation. Scatterometers, active microwave instruments, are capable of retrieving wind speed under rainy conditions and their data are used in conjunction with passive wind observations by several of the retrieval algorithms presented in the manuscript (see Section 2). For more details of wind speed data used in the various E (LHF) retrievals, we refer the referee to the literature cited in Section 2.1.
[Meissner, T. and Wentz, F.J.: The Emissivity of the Ocean Surface Between 6 and 90 GHz Over a Large Range of Wind Speeds and Earth Incidence Angles, IEEE Trans. Geosci. Rem. Sens., 50(8), doi: 10.1109/TGRS.2011.2179662, 2012.]

*In addition, If E estimates are based on BT measurements, which are more accurate in clear sky conditions than in cloudy sky conditions, wouldn't that cause a bias? How is it calculated in cloudy (and rainy) conditions? If it is only calculated in clear sky conditions, wouldn't the E estimations be biased high (as in cloudy and rainy conditions the evaporation is lower)?*

The retrieval of E, LHF, and wind speed from passive microwave observations is not possible in scenes with heavy precipitation. This is, indeed, expected to lead to a positive bias, but it has not been quantified. Interestingly, global mean E from ERA5 exceeds all satellite-based estimates, despite the fact that all sky conditions are included. We inserted the following statement after line 527 (Section 5): "It is interesting to note that satellite-based E are very likely biased high by the removal of scenes with strong precipitation (where the retrieval of WS, LHF, and E is not possible). In this light, the difference in E between ERA5 and the satellite-based retrievals should actually be larger than observed in Fig.3, as monthly mean E is determined from all sky conditions in reanalysis. As the OAFlux and SEAFLUX blend satellite estimates with continuous background fields (Sect. 3), these algorithms should be less impacted by such sampling biases."

*L368: is the largest deviation in E estimations in the tropics due to the large (and optically thick) cloud cover?*

Passive microwave instruments "observe" humidity in the total column with only limited information on the vertical structure. Retrievals of near-surface humidity are most accurate under typical conditions of moisture stratification. Biases arise when the vertical stratification of moisture departs strongly from these typical conditions. Thus, estimates of near-surface humidity and, subsequently, E often vary between products (Roberts et al., 2019). Note that the most recent J-OFURO and SEAFLUX products include additional a priori information on moisture stratification within the retrieval algorithms to mitigate these issues. Accounting for this improves the consistency of retrieval results appreciably compared to *in situ* measurements (Roberts et al., 2019).

We added this information to the manuscript by modifying line 368 to:

"(…) while the largest deviations appear mainly in the tropics. This is due to the frequent occurrence of weather conditions in which the moisture stratification departs substantially from typical conditions to which the retrieval algorithms of near-surface moisture are tuned. Accounting for this dependence on moisture stratification, as in the SEAFLUX and J-OFURO algorithms, improves retrieval results appreciably compared to *in situ* measurements (Roberts et al., 2019)."

*L437: I think that the correlation does not decrease when Delat Qocean is not considered because there is basically no correlation even when it is considered. So, it can't get any lower than that. Is that correct?*

In principle, the referee is correct. However, the statement in the manuscript is not quite accurate, as there is appreciable improvement in $R^2$ for J-OFURO-G and IFREMER-G for yearly means and monthly anomalies. With the updated SEAFLUX3 statistics, Table 3 reads:

| Data set | Monthly mean | Yearly mean | Monthly anomaly |
|---|---|---|---|
| HOAPS-4.0 | 0.03 | 0.00* | 0.06 |
| J-OFURO3 - GPCP-1DD | 0.16 | 0.31 | 0.22 |
| IFREMER4.1 - GPCP-1DD | 0.13 | 0.23 | 0.20 |
| OAFlux3 - GPCP-1DD | 0.14 | 0.01* | 0.11 |
| SEAFLUX3 - GPCP-1DD | 0.17 | 0.02* | 0.12 |
| ERA5 | 0.86 | 0.86 | 0.83 |

Ignoring the contribution of $\nabla(vq)$ yields:

| Data set | Monthly mean | Yearly mean | Monthly anomaly |
|---|---|---|---|
| HOAPS-4.0 | 0.01* | 0.00* | 0.01* |
| J-OFURO3 - GPCP-1DD | 0.12 | 0.19* | 0.08 |
| IFREMER4.1 - GPCP-1DD | 0.12 | 0.17* | 0.08 |

| OAFlux3 - GPCP-1DD | 0.11 | 0.00* | 0.01* |
|---|---|---|---|
| SEAFLUX3 - GPCP-1DD | 0.11 | 0.00* | 0.00* |
| ERA5 | 0.42 | 0.57 | 0.55 |

Hence, including the $\nabla(vq)$ contribution not only improves correlations of yearly means and monthly anomalies, but also yields more cases where the correlation is significant (not marked by an asterisk). We corrected the statement, while also changing the notation of $\nabla Q$ to $\nabla(vq)$, as recommended by referee #2.

"Including the contribution of $\nabla(vq)$ improves the correlation appreciably for ERA5, as mentioned above. For satellite data the correlation also improves, particularly for yearly means and monthly anomalies of IFREMER-G and J-OFURO-G (not shown)."

*Technical comments:*
*L208: us–>use*
Corrected.

*You alter between italic and non-italic in P, E and E-P. I think it should all be italic.*
In accordance with convention, all symbols are written in italic. The fact that the abbreviations chosen for evaporation, precipitation, and freshwater flux are the same as their respective symbols (see Table 2) causes E, P, and E-P to be written in italic in some cases and non-italic in others. This should not lead to confusion, so we would like to keep the notation this way in the paper.

*References*
*[1] Allan, R. P. et al. Advances in understanding large-scale responses of the water cycle to climate change. Annals of the New York Academy of Sciences (2020).*
*[2] Dagan, G., Stier, P. & Watson-Parris, D. Analysis of the atmospheric water budget for elucidating the spatial scale of precipitation changes under climate change. Geophysical Research Letters (2019).*
*[3] Yin, J. & Porporato, A. Looking up or looking down? Hydrologic and atmospheric perspectives on precipitation and evaporation variability. Geophysical Research Letters 46, 11968-11971 (2019).*
*[4] Liepert, B.G. & Previdi, M. Inter-model variability and biases of the global water cycle in CMIP3 coupled climate models. Environmental Research Letters 7, 014006 (2012).*
*[5] Knutti, R. & Sedláček, J. Robustness and uncertainties in the new CMIP5 climate model projections. Nature Climate Change 3, 369 (2013).*
*[6] Held, I. M. & Soden, B. J. Robust responses of the hydrological cycle to global warming. Journal of Climate 19, 5686-5699 (2006).*

We thank the referee for pointing us to these papers and have incorporated them into the revised manuscript. In particular, we added the global total fluxes estimated in [1] to Table 4. Please note that [6] is already cited in the first version of the manuscript.

The updated Table 4:

**Table 4.** Estimates of ocean total $E$ and $P$, land and ocean total $E - P$, net transport of water vapor, and continental runoff given in $10^3$ km$^3$ yr$^{-1}$. The upper three rows contain results from this study, the lower four those from earlier investigations. ERA5 estimates are calculated from ensemble mean data, the standard deviation (std) is derived from ensemble statistics. The satellite-based data sets used in our study were averaged to obtain the mean and std of observed (Obs.) $E_{ocean}$ and $P_{ocean}$, and the range is given in the third row. Net water vapor flux divergence over land ($\nabla Q_{land}$) and ocean ($\nabla Q_{ocean}$) and continental runoff $R$ are given in the last three columns. The estimates from the study by Rodell et al. (2015) are separated into observations (obs.) and model-optimized observations (opt.), see the text for details.

| | $E_{ocean}$ | $P_{ocean}$ | $(E-P)_{ocean}$ | $(E-P)_{land}$ | $\nabla Q_{land}$ | $\nabla Q_{ocean}$ | $R$ |
|---|---|---|---|---|---|---|---|
| ERA5 | $467 \pm 1$ | $426 \pm 2$ | $43 \pm 2$ | $-44 \pm 0.4$ | $-43 \pm 0.2$ | $31 \pm 0.2$ | 42.1 |
| Obs. mean $\pm$ std | $425 \pm 20$ | $360 \pm 25$ | $52 \pm 13$ | $-$ | $-$ | $-$ | $-$ |
| Obs. range | 397–453 | 335–384 | 35–65 | $-$ | $-$ | $-$ | $-$ |
| Oki and Kanae (2006) | 436.5 | 391 | 45.5 | $-45.5$ | $-45.5$ | 45.5 | 45.5 |
| Trenberth and Asrar (2014) | 413 | 373 | 40 | $-40$ | $-40$ | 40 | 40 |
| Rodell et al. (2015) obs. | $410 \pm 36$ | $385 \pm 39$ | $25 \pm 53$ | $-45 \pm 9$ | $-43 \pm 8$ | $47 \pm 19$ | $50 \pm 7$ |
| Rodell et al. (2015) opt. | $450 \pm 22$ | $403 \pm 22$ | $47 \pm 31$ | $-46 \pm 7$ | $-46 \pm 4$ | $46 \pm 2$ | $46 \pm 4$ |
| Allan et al. (2020) | $480 \pm 48$ | $434 \pm 43$ | $46 \pm 65$ | $-46 \pm 14$ | $-46 \pm 5$ | $46 \pm 5$ | $51 \pm 3$ |

And we added the following lines to Sect. 4.6:

"The global total fluxes estimated by Allan et al. (2020) derive from Rodell et al. (2015), but following the recommendation by Stephens et al. (2012), $E_{ocean}$ and $P_{ocean}$ were both increased by 30·10$^3$ km$^3$ yr$^{-1}$ to improve the agreement with energy constraints, yet keeping land-ocean fluxes constant. These increases are larger than the ±22·10$^3$ km$^3$ yr$^{-1}$ uncertainty on $E_{ocean}$ and $P_{ocean}$ estimated by Rodell et al. (2015) based on the optimized method and so a more modest increase of 20·10$^3$ may be appropriate. This would produce fluxes of $E_{ocean}$ = 470·10$^3$ km$^3$ yr$^{-1}$ and $P_{ocean}$ = 424·10$^3$ km$^3$ yr$^{-1}$ that are quite close to ERA5 estimates (R. Allan, personal communication, Oct. 2020)."

---

## Author Comment (AC2) · 21 Oct 2020

Authors' reply to Anonymous Referee #2

We thank the referee for his/her constructive and detailed review of our paper. We have adapted the manuscript in accordance with the referee's suggestions, as detailed below. Referee comments appear in *italics*, our reply in normal font, and changes to the manuscript in blue.

*Review of "Intercomparison of freshwater fluxes over ocean and investigations into water budget closure" by M. Gutenstein et al.*
*This is a nice intercomparison study of various satellite-based precipitation and evaporation products and ERA5. The authors show that there is a large spread among the different products, and most of them fail to satisfy global budget constraints, especially when combining different products for estimation of P, E, and the transports. ERA5 performs best with a remarkably good agreement of forecast-based fluxes (P and E) and analysis-based transports (moisture flux divergence). The paper is well-structured and -written and therefore easy to follow.*
Thank you!

*My only major comment is on the obvious error in the computation of ERA5 moisture transports as detailed below.*
*Moreover, the reader would probably like to see stronger conclusions. I know it is hard to make a ranking, but e.g. SEAFLUX with its clearly unphysical P-E over ocean could be ruled out as clearly unrealistic. Also HOAPS appears to be a bit of an outlier, especially in terms of variability. The budget constraints are an objective measure to rule out poor data, and this helps to better constrain the best estimate of the water budget, without unnecessary inflation of the error bars. Otherwise I only have a number of minor comments.*
We would certainly have liked to conclude with more clear recommendations, but believe that, apart from the finding that the global total E from SEAFLUX2 is unrealistic, our results do not permit any kind of ranking. First, there are not enough truly independent data with which to assess the quality of each data set. For example, although HOAPS precipitation is an outlier compared to other data sets, it is not certain that its variability (e.g. dependence on ENSO) is erroneously large, as there are reasons to believe that ERA5 and GPCP underestimate variability. This, however, is the topic of a future study. Second, each data set has its particular strengths and weaknesses, and HOAPS comes closer to water budget closure than OAFLUX or IFREMER. For these reasons, our conclusions cannot go beyond the statement that observational data sets (and associated uncertainty estimates) need to improve.
We added this information to Section 5: "Although it is tempting to make a ranking from the results of our inter-comparison, there are good reasons to resist. First, there are not enough truly independent data with which to objectively assess the quality of each data set. And second, each has its particular strengths and weaknesses: for example, HOAPS comes closer to water budget closure than OAFLUX or IFREMER (panel c of Fig. 3)"

And we modified our final statement as follows: "In general, the quality of observations of the water cycle needs to improve before attempts at assessing effects of climate change from those data can be undertaken. The importance of accompanying high-quality uncertainty information cannot be overstated."

*Major comment: The authors use VIMD from ERA5 to compute ocean-to-land moisture transports. For this it should not matter whether on integrates VIMD over all land points or over all ocean points. Another constraint is that the global average of VIMD must be zero. This is a mathematical constraint independent of data quality. However, the authors obtain inconsistent results for land and ocean integrals of VIMD (table 4). So either the archived VIMD fields are flawed (which I doubt as I am using the ERA5 data myself and could not find a similar problem) or there is some error in the author's*

*processing chain that leads to these erroneous results. If this problem is really due to interpolation errors, as suspected by the authors, these interpolation errors are clearly unacceptably large. In short, this error must be corrected.*

We could not agree more with the referee: summed over the globe, VIMD should equal 0. However, two of us calculated the area-weighted sum over the globe in three different ways and every time we find the rather constant value of -0.04 mm/day. These calculations were performed with monthly mean VIMD, regridded at ECMWF to a regular, 1x1 degree grid. This puzzled us and we contacted the ERA5 team about it at the beginning of 2020. Paul Berrisford gave us the explanation that we eventually wrote into the manuscript, i.e., the error occurs during transformation from the model grid to a regular lat/lon grid. Following the referee's comment, we contacted the ERA5 team again. Paul Berrisford confirmed the finding that global mean ERA5 VIMD equals -0.04 mm/day and commented: "Anton [Beljaars] also points out: 0.04 mm/day is not very big. I do not think accuracy at that level can be claimed on the fields of the water cycle. Also observationally, I doubt whether global precipitation is known to within one tenth of a mm/day."

Whereas this is certainly true for a single grid cell, we, like the referee, feel that at the global scale 0.04 mm/day (which sums up to about $10 \cdot 10^3$ km$^3$/year, as seen in our Table 4) is large – perhaps even unacceptably large. However, a detailed discussion of the issue is not within the scope of our manuscript and should take place elsewhere, preferably with direct involvement of the ERA5 experts. To address this issue more clearly, we changed lines 413-415 to: "However, we find global total ERA5 VIMD to be -0.04 mm d$^{-1}$: a small value within the standard deviation of the ensemble of single grid boxes, but significant and on the order of the amplitude of the seasonal cycle of net E-P on the global scale. The deviation from zero is due to the fact that VIMD is calculated in grid point space (and not in the model's spectral space), where the mathematical constraint of net zero divergence is not enforced (P. Berrisford, personal communication, Oct. 2020)."

And line 497-499 to: "As observed above, the fact that ERA5 VIMD is calculated in grid point space causes $\nabla(vq)$ to be about $10 \cdot 10^3$ km$^3$ yr$^{-1}$, and not zero. In addition, due to the tighter observational control over land, analysis increments may be larger over ocean than over land and may cause $\nabla(vq)$ to be very close to net *E-P* over land, but less so over ocean (P. Berrisford, pers. comm. Oct. 2020). "

*Minor comments:*
*Equation 1 and everywhere else: as it stands, the VIMD terms looks like the moisture gradient. I suggest to replace with the more appropriate nabla * (vQ).*
The referee is right. We replaced the inaccurate term with $\nabla(vq)$ in the equations and text.

*L17: I presume you use monthly values. Please say it clearly, as the correlation strongly depends on the considered timescales.*
Correct. We changed the sentence to reflect this:
"On a monthly time scale, linear regression of $E_{ocean} - \nabla(vq)_{ocean}$ with $P_{ocean}$ yields $R^2 = 0.86...$"

*L37: The term "model reanalysis" seems an uncommon term to me. I suggest to drop "model". If you want to give an attribute, it may be better to say "climate reanalysis" or "dynamical reanalysis".*
Done.

*L46: Isn't there an author on the GPCP document?*
This is not a document, but the DOI reference to the data set itself; no single author is indicated. In fact, GPCP-1DD v.1.3 was not correctly cited in the manuscript, and we updated it accordingly:
"Mesoscale Atmospheric Processes Branch/Laboratory for Atmospheres/Earth Sciences Division/Science and Exploration Directorate/Goddard Space Flight Center/NASA, and Earth System Science Interdisciplinary Center/University of Maryland: GPCP Version 1.3 One-Degree Daily Precipitation Data Set, Research Data Archive at the National Center for Atmospheric Research, Computational and Information Systems Laboratory, doi: 10.5065/PV8B-HV76, 2018. Accessed June 2019."

*L58: "moisture divergence" is sloppy terminology. Moisture itself cannot diverge. It should be "moisture flux divergence".*
Corrected.

*L58: another nitpicky comment: VIMD is technically not identical with advection, although it is an excellent approximation of it. I suggest a slightly more cautious wording.*
Corrected.

*L85-86: "model runs" sounds it does not use any observational info. Simply say it consists of ten ensemble members.*
Corrected.

*L108: Le is usually called "latent heat of evaporation".*
Corrected.

*L133: Do you mean SST averaged over the top 0.5m? Please clarify.*
We wrote: "(...) a bulk SST at 0.5 m", which refers to the minimum depth at which the sondes measure to which satellite data are calibrated. This was clarified in the manuscript by changing the text to: "(...) the SST at a depth of 0.5 m".

*L183-184: The statement about forecast skill is hard to understand for a non-expert.*
Agreed, but it is not of importance for the rest of the study, so we will not explain it further here.

*L184: "model runs" -> see comment above.*
Corrected.

*L184: It should be mentioned that the ERA5 ensemble members have a lower resolution than the stated 30km. How would the results change when using the high-resolution ERA5 data?*
The referee might have misunderstood that we did, indeed, perform the study with high-resolution data (although those were interpolated to a regular 1 degree grid by ECMWF prior to our investigations). Only uncertainty estimates were made using (lower-resolution) ensemble data that was similarly interpolated to a 1 degree grid. The text was modified to make this more clear: "ERA5 encompasses data from ten reanalysis runs at a reduced spatial resolution of 62 km, allowing estimation of the uncertainty range from ensemble statistics. The analysis presented here is performed with the ECMWF ensemble mean, whereas uncertainty is determined from the ensemble. Both data sets were interpolated to 1° resolution at ECMWF."

*L189: I suggest to delete "on single levels"*
Done.

*L192: It should be noted that monthly P and E from ERA5 is averaged from short-term forecasts (12 or 24 hours? Needs to be clarified as well!)*
We changed the text on lines 191-195 to: "Monthly averages are calculated from daily means starting at 00 UTC and ending at 00 UTC the following day (ECMWF, 2020). Evaporation rates are derived from the gradients of specific humidity between the surface and the lowest model level (10 m for ERA5) as described above (ECMWF, 2016). The main differences between the satellite-based retrievals described here and ERA5 determination of $E$ are the consistency of atmospheric variables involved ($u$, $q_a$, $q_s$) and the high temporal sampling rate: monthly means are determined from (daily means of) hourly data from forecasts initialized daily at 6:00 and 18:00 UTC."

*L198: If the TCWV tendency is computed from monthly means (rather than instantaneous values at beginning and end of the month), one should use centered differences.*

The referee is right. We re-computed TCWV tendencies as suggested and updated Fig. 7. The figure changed only slightly:

[Figure]

Figure 7. ERA5 monthly mean E-P over the whole globe (black), land only (green), and ocean (blue); global mean ΔW (light blue), mean ∇(vq) over land (pink) and ocean (purple). The mean values over the globe and land were scaled by their surface area relative to the ocean surface area (i.e., they were multiplied by 510/350 and 160/350, respectively) to obtain consistency with the over-ocean means shown in Fig. 3. Error bars represent the standard deviation within the 10-member ensemble, which is smaller than the graph's line width for E-P over land, ΔW, and ∇(vq).

Line 198 now reads: "We calculated the TCWV tendency in month $x$ from monthly mean ERA5 data by subtracting TCWV of month $x$+1 from TCWV of month $x$-1, then dividing by 30 days/month (…)"

*L243: When deriving E from monthly Q fields, what is the error from neglect of sub-monthly covariance between E and SST?*
A test with HOAPS-4.0 data shows that the error due to the derivation of E from monthly LHF and SST is systematic, but of negligible magnitude. This is shown in the figure below, which depicts the differences between E derived from monthly means and monthly mean E from instantaneous values. Monthly mean difference maps are very similar to the left panel below, with the mean global difference 0.005 mm d$^{-1}$ and grid point differences barely exceeding 0.01 mm d$^{-1}$.

[Figure]

Figure A. Difference between E derived from monthly mean HOAPS LHF and SST and monthly mean HOAPS E for the time period 1997-2013. Left, climatological difference; right, root mean square differences.

We included this information into the manuscript by inserting the following statement on line 248: "Applying the same method of calculating E from HOAPS monthly mean LHF and SST data causes

negligible differences with monthly mean E determined from instantaneous LHF and SST data (root mean square differences of ≤0.01 mm d$^{-1}$ for individual grid boxes during 1997-2013)."

*L258: I suggest to replace "relative" with "area-specific" and "total" with "area-integrated".*
Done.

*L260: Please clarify how sea ice is treated in general. I presume it is masked out? Is this a seasonally varying mask?*
As mentioned in Section 3, we neglected the seasonally changing number of observations screened out by the sea ice mask in our study. We clarified this by changing the statement on line 260 to:
"Seasonally varying numbers of observations screened out due to sea ice are neglected."

*L273: "at the ITCZ" maybe better "in the ITCZ"?*
Corrected.

*L280: Figure 2: One could make the simple statement that HOAPS differences are generally larger (RMS values of the field would be useful), but areas of disagreement are smaller because of the larger uncertainties.*
That is an excellent suggestion. We added the total RMSD to the text and inserted a statement similar to the one suggested by the referee to line 280:
"(...) with collocated ERA5 data. Although HOAPS differences with ERA5 appear larger to the eye, the root mean squared (RMS) differences are 0.6 mm d$^{-1}$ for each of the three comparisons: 0.60 mm d$^{-1}$ for HOAPS, 0.58 mm d$^{-1}$ for SEAFLUX, and 0.57 mm d$^{-1}$ for OAFlux. As already seen in Fig.1, differences are not homogeneously distributed over the globe."

*L285: similarly -> similar*
We changed the word to the more appropriate also.

*L317: In terms of flow of reading, it may be better to move the sentence about ENSO correlation further down to around L340.*
Done.

*Figure 3 and in general: I suggest to change the panel labelling to small letters, as capital letters have potential for confusion, especially "E". Best would be E -> (e)*
Panel labelling was changed to small letters.

*Figure 3: It would be interesting to see the ENSO correlation for every curve. This could be given in the legend, ideally with the lag at which the maximum correlation occurs.*
Good suggestion, we updated the figure with correlation coefficients and corresponding time lag. We also improved the readability of the left panels of Fig.3 by changing the depiction of the uncertainty information.

[Figure]

Figure 3. Climatological (1997--2013) seasonal cycle of global ocean mean evaporation rate (a), precipitation rate (b), and freshwater flux (c). HOAPS, ERA5, OAFlux, SEAFLUX, and GPCP yearly mean values and associated 1σ uncertainty ranges are shown in the boxes to the right of the panels. Monthly mean anomaly (w.r.t. the climatological seasonal cycle depicted at left) over the global oceans (80°S--80°N) of evaporation rate (d), precipitation rate (e), and freshwater flux (f). The anomaly data are smoothed using a three-month running mean. Panel e additionally displays the Niño3.4 index shifted by +3 months (right y-axis). The legend additionally displays the correlation coefficient of the Niño3.4 index with P anomalies and the time lag of highest correlation (Δt in months). Ticks on the time axis mark January of the indicated year.

*L319: Is "bias" the right term? We see differences, but still one of the datasets could be unbiased.*
The referee is right. We replaced the word "bias" in the text with "difference" or "deviation".

*L337: "biased low". SEAFLUX seems to be low in general (according to the mean annual cycle figure). So better to change to something like "particularly low".*
The statement was changed completely after updating the SEAFLUX data to version 3.

*L373: Is there a reference for the statement on detection of snow in HOAPS?*
The remark was actually meant generally for all global satellite-based precipitation data sets (or at least the two used for the present study). In an inter-comparison study, Tapiador et al. (2017) show that compared to satellite-borne radar observations, GPCP detects too little precipitation in the higher latitudes. HOAPS performs slightly better, but still underestimates precipitation near the poles. We added this information to the manuscript by changing line 373 to:
"(…) and in part to difficulties pertaining to the detection of snow by passive microwave instruments (Tapiador et al., 2017; Kidd and Huffman, 2012)."

*L389: remove "are"*
Done.

*L413: This statement would be correct if VIMD was the 3D-divergence, i.e. including fluxes at top of the atmosphere, where there theoretically could be an exchange with space. However, your VIMD is 2D and its global average is 0 according to the sentence of Gauss.*
Please see our answer to the first major comment above.

*L415: See my major comment.*
Please see our answer to the first major comment above.

*L503: I think it should be "right-most"*
Absolutely. We corrected the error.

*L575-576: Please provide a reference for this statement.*
We corrected the estimates to 2%-3% K$^{-1}$ and added a reference to a paper recommended by referee #1: Allan, R. P. et al. Advances in understanding large-scale responses of the water cycle to climate change. Annals of the New York Academy of Sciences (2020).

*L586: This statement is on "observation-based attempts". Please clarify.*
The referee is right: as it is, the statement is inaccurate. As mentioned above, we modified it to: "In general, the quality of observations of the water cycle needs to improve before attempts at assessing effects of climate change from those data can be undertaken. The importance of accompanying high-quality uncertainty information cannot be overstated."

*Figure 4: middle and right columns: Would it be possible to use color schemes that are really white in the middle?*
It is possible to use a different color scale, but we would like to keep gray as the center color to distinguish from missing values, which we prefer to show in white.

*Table 4: How is runoff from ERA5 obtained? Is this the area-integral of all grid point values?*
Yes, it is. We clarified this in the manuscript in Sect. 3: "Global total runoff from ERA5 and other data sets was determined by calculating the area integral of all points."

---

## Author Comment (AC3) · 21 Oct 2020

Authors' reply to Anonymous Referee #3

We are very grateful to the referee for his/her appreciation of the effort that went into the creation of our manuscript and we thank him/her for the detailed comments to our manuscript. We have done our best to answer all questions and incorporate the referee's suggestions. In our reply below, referee comments appear in *italics*, our reply in normal font, and changes to the manuscript in blue.

*General Comments:*
*It is an overall well-written manuscript comprehensively addressing a truly important subject on freshwater fluxes over ocean and (the associated) water budget closure by using and intercomparing seven credible and lengthy globally-covered products of evaporation and/or precipitation (six generally satellite-based datasets, along with one reanalysis). This reviewer would like to acknowledge the authors for their willing to challenge themselves and tackle such a subject that would almost (if not absolutely) guarantee and require tremendous efforts and time, and persistent commitments, let alone equipping with solid expertise and knowledge, and great ideas and insights. However, a few relatively major concerns may need further elaboration or revisions. Several minor revisions are also suggested.*
*Finally, this reviewer considers that the authors deserve a solid credit, again for their tremendous efforts and crucial works. This reviewer will highly recommend this manuscript for publication by Hydrology and Earth System Sciences (HESS) (an Open Access Journal) should the following suggested revisions be properly conducted accordingly.*
We are much obliged!

*Major Comments/Suggestions/Revisions:*
*1) E-P uncertainty involving E uncertainty and P uncertainty:*
*The uncertainties of E-P of different sets of products involved and targeted in this study should mainly be depending on the respective uncertainties of the P and E products, e.g., for IFREMERG(E-P) = IFREMER(E) - GPCP(P), the uncertainty of IFREMER-G(E-P) should supposedly be an added sum of the uncertainty of IFREMER(E) and the uncertainty of GPCP(P). These respective E and P uncertainties, however, are not among the main focuses of this study (as the authors have indicated). It'd be very understanding and foreseeable that such (additional) tasks of investigating and analyzing (or may even need to newly generate or estimate) the respective P and E uncertainties would add up another level of difficulty and efforts, especially if/when researchers were not directly involved in those P and E productions, and the currently available related uncertainty info's have been quite limited (which have also been revealed in this manuscript). As for "We conclude that for a better understanding of the global water budget, the quality of E and P data sets themselves and their associated uncertainties need to be further investigated", this reviewer has fully agreed on this critical "conclusion" finding, which, honestly speaking, has also been "expected" during the midst of review. It might also be fair and reasonable to alternately say "for a better understanding of the global fresh water (E-P or P-E triggered by seeing the assumptions made in Eqs. 1-3)*
The referee is correct in stating that the total E-P uncertainty is the sum of the uncertainty of both E and P components. Or, to be even more exact: the root of the squared sums. That is how we calculated total uncertainty ranges for HOAPS and OAFLUX-G (as described in Sect. 3). At the time of first submission, none of the other E (or LHF) satellite data sets contained uncertainty estimates. The recent publication of SEAFLUX version 3 allowed us to update our manuscript with much improved SEAFLUX E data. As these contain associated random uncertainty estimates, we updated Fig.2, where previously only HOAPS and OAFLUX-G were examined for significant differences with ERA5, with SEAFLUX-G data:

[Figure]

Figure 2. Left panels: Difference maps of HOAPS (upper), OAFlux-G (center), and SEAFLUX-G (lower) climatological mean E-P minus the corresponding collocated ERA5 climatology (1997-2013). Right panels: HOAPS (upper), OAFlux-G (center), and SEAFLUX-G (lower) climatological mean 1σ uncertainty. White lines in the left panels enclose regions where the difference with ERA5 E-P exceeds the 2σ uncertainty range.

Moreover, we changed the word "investigated" on line 23 to "improved", and the statement on line 568 was changed, following the suggestions of referees 2 and 3:

"In general, the quality of observations of the water cycle needs to improve before attempts at assessing effects of climate change from those data can be undertaken. The importance of accompanying high-quality uncertainty information cannot be overstated."

*2) The assumptions of neglecting $\nabla Q$ or $\Delta W$ for global or regional scales have trigged this reviewer wondering about their potential impact on the E-P uncertainty. Here's the comment that the authors may feel free to respond or not (Optional).*

*In the case of omitting $\nabla Q$ or $\Delta W$, it might cause two folds of potential impacts, hypothetically: 1) if neither $\nabla Q$ nor $\Delta W$ would carry uncertainties, then the currently estimated uncertainty of EP in this study could have been overestimated since part of the estimated uncertainty might have been implicitly contributed by the being-omitted "true" amounts of $\nabla Q$ or $\Delta W$ (even though being small, but had been neglected), 2) if either/both $\nabla Q$ or/and $\Delta W$ would carry uncertainties, then the currently estimated uncertainty of E-P in this study could have been underestimated, especially if/since the uncertainties of $\nabla Q$ or $\Delta W$ (if available and "certain") would be sufficiently large (greater than their true amounts) and add an additional net amount of uncertainty into E-P, otherwise. The E-P*

*uncertainty could have still been overestimated as in "1)" if the true amounts of ∇Q or ΔW are larger than their respective uncertainties.*

To answer this interesting comment, we need to separate the treatment of $\nabla Q$ (corrected to $\nabla(vq)$ in in the revised manuscript) and $\Delta W$. The global total of $\nabla(vq)$ is exactly 0, hence Eq. 2 is exact on a global scale and not an approximation (please note that the non-zero global total $\nabla(vq)$ found in ERA5 was commented upon by Referee #2). This is not the case for $\Delta W$, which we chose to ignore on regional scale for time periods larger than one month. Averaged over the globe, $\Delta W$ varies between -0.03 and 0.03 mm d$^{-1}$ , balancing the global total E-P (Eq. 2 and Fig. 7 in the manuscript). Regionally, monthly $\Delta W$ are at least a factor of 10 smaller than E-P (or $\nabla(vq)$), as shown exemplarily for December 2013 below.

[Figure]

Figure A. Maps of TCWV tendency, VIMD, and E-P from ERA5 for November 2013. Note that the color scale is a factor of 10 smaller for TCWV tendency than for VIMD or E-P.

On a monthly and regional scale, $\Delta W$ and the associated uncertainty (which is small) can be safely ignored and $\nabla(vq)$ should be equal to E-P. This is very nearly exactly true for ERA5, as can be seen in the figure above, by the high correlations noted for ERA5 in Table 3 and in Fig. A2 in the manuscript. On lines 390—392, we note that: "It can, however, be argued that VIMD from reanalysis is a more reliable quantity than reanalysis E-P, since VIMD is calculated from the state variables wind and water vapor, whereas E and P are model-physics derived (e.g., Trenberth et al., 2011))." The uncertainty of VIMD is, therefore, smaller than that of E-P, but of its magnitude we can only say that the uncertainty range determined from the ensemble is, in all likelihood, a lower bound.

The correlation coefficients in Table 3 were calculated without taking into account uncertainties, and the uncertainty ranges for Table 4 were derived from the E, P, E-P and other data sets themselves.

Hence, we appreciate the thoughts of the referee on the neglect of terms ($\Delta W$ and $\nabla(vq)$) as an additional source of uncertainty to budget calculations, but do not see at which point in the manuscript this may be relevant.

*3) On line 272 (related to Fig. 1):*
*A seemingly correct but actually questionable (not quite proper) statement that needs a "major" attention and revision. "A and B shows that the E-P pattern is mainly determined by P, as there is less spatial variation in E."*
*It should be more accurate to state "A and B shows that the E-P pattern is mainly determined by P in the tropical and high-latitude regions, but determined by E in the subtropical regions." It's also the absolute magnitudes/intensities of E or P, not just only their spatial variation that would matter. Actually, if "warm" color (red) and "cold" color (blue) were applied for E (panels C and D) and P (panels E and F), respectively, such crucial features (reviewer's points) would have been better revealed. Perhaps, the authors may genuinely consider it.*

The reviewer is right about the statement on line 272, and we corrected it as suggested. We also modified the color scale of the evaporation and precipitation plots in Fig. 1:

[Figure]

Figure 1. Satellite ensemble median (SEM) and ERA5 climatologies (1997-2013) of freshwater flux (a and b) and evaporation (c and d), and GPCP and ERA5 precipitation (panels e and f). ERA5 data coverage was reduced to match satellite data (see text for details).

*4) The authors would sometimes describe, elaborate or discuss the features or findings shown via figures or panels, but without consistently and explicitly citing them along the presentation/writing.*
The referee is right. We added references to the respective figure panels on lines 308, 325, and 337.

*Minor Revisions:*
*1) On line 46:*
*Suggest change "are the most widely used data sets." -> "are among the most widely used data sets."*
Done.

*2) On lines 58-59:*
*Suggest change "With W the total column water vapor and $\nabla Q$ the moisture divergence, i.e., the amount of moisture removed by advection from the considered volume." -> "With W the total column water vapor and $\nabla Q$ the total moisture divergence or convergence, i.e., the amount of moisture associated with the moisture advection and the mass divergence or convergence scaled by water vapor from the considered volume."*

The statement on lines 58-59 was not quite accurate. We replaced it with: "With W the total column water vapor and ∇(qv) the moisture flux divergence, i.e., the amount of moisture removed by dynamical transport from the considered volume."

*3) On line 157:*
*Suggest change "Version 2 (GSSTF2.0, Shie et al. (2009))" -> "Version 2; 2c (GSSTF2.0, Chou et al.*
*(2003); GSSTF2c, Shie et al. (2009))" and the following respective reference should thus be included:*
*Chou, S.-H., E. Nelkin, J. Ardizzone, R. Atlas, and C.-L. Shie, 2003: Surface Turbulent Heat and*
*Momentum Fluxes over Global Oceans Based on the Goddard Satellite Retrievals, Version 2*
*(GSSTF2), J. Climate, 16, 3256-3273. [10.1175/1520-0442(2003)016<3256:STHAMF>2.0.CO;2]*
Done.

*4) On line 206:*
*Suggest revise "see, e.g., Kidd and Huffman, 2011; Tapiador et al. , 2017)" -> "see, e.g., Kidd and*
*Huffman, 2011; Tapiador et al., 2017)"*
The spurious space is caused by the HESS LaTex template. Hopefully, the copy editor will remove it before publication.

*5) On lines 207-208 :*
*Suggest revise "which makes us of its own P data)" -> "which makes use of its own P data)"*
Done.

*6) On line 210:*
*Suggest change "The Global Precipitation Climatology Project - 1 Degree Daily (GPCP-1DD; denoted*
*GPCP hereafter)" -> "The Global Precipitation Climatology Project - 1 Degree Daily (denoted GPCP and*
*GPCP-1DD, respectively, hereafter)"*
In the presented study, only GPCP-1DD data are used for comparison and for calculation of E-P. In an attempt to make the manuscript, which contains many acronyms, more readable, we left out the versions of the algorithms in the main text (e.g., HOAPS-4.0 is referred to as "HOAPS"). For brevity and consistency, we prefer to denote GPCP-1DD as "GPCP" throughout the manuscript.

*7) On line 235:*
*Suggest revise "consisting of 10 seperate model runs" -> "consisting of 10 separate model runs"*
Following the advice of referees 1 and 3, the sentence was revised to: "consisting of 10 separate reanalysis runs".

*8) On line 389:*
*Suggest revise "we are also examine the separate contributions" -> "we also examine the separate*
*contributions"*
Done.

*9) On line 458:*
*Suggest revise "(Oki and Kanae , 2006 " -> "(Oki and Kanae, 2006;"*
Please see our answer to comment 4.

*10) On line 574:*
*Suggest revise "(e.g, Allen and Ingram , 2002; Held and Soden , 2006; Trenberth et al., 2007)." ->*
*"(e.g., Allen and Ingram, 2002; Held and Soden, 2006; Trenberth et al., 2007)."*
Again, please see our answer to comment 4.

*11) In Fig. 3B:*
*There are four kinds of shading shown with three precipitation data sets. The extra "dummy" shading*
*should be removed.*

This is not dummy shading, but the overlap of two colors. This was apparently somewhat confusing and the addition of the SEAFLUX3 uncertainty range did not improve the readability of the plot, therefore, we changed Fig. 3 and its caption to:

[Figure]

Figure 3. Climatological (1997--2013) seasonal cycle of global ocean mean evaporation rate (a), precipitation rate (b), and freshwater flux (c). HOAPS, ERA5, OAFlux, SEAFLUX, and GPCP mean values and associated 1σ uncertainty ranges are shown in the boxes to the right of the panels. Monthly mean anomaly (w.r.t. the climatological seasonal cycle depicted at left) over the global oceans (80°S--80°N) of evaporation rate (d), precipitation rate (e), and freshwater flux (f). The anomaly data are smoothed using a three-month running mean. Panel e additionally displays the Niño3.4 index shifted by +3 months (right y-axis). The legend additionally displays the correlation coefficient of the Niño3.4 index with P anomalies and the time lag of highest correlation (Δt in months). Ticks on the time axis mark January of the indicated year.

*12) On line 573-574:*
*"its water-holding capacity increases at a rate consistent with the Clausius-Clapeyron relationship (e.g, Allen and Ingram , 2002; Held and Soden , 2006; Trenberth et al., 2007)"*
*Here's one paper (i.e., Shie et al., 2006) that also addressed a Clausius-Clapeyron scenario focusing on tropical oceans. It may be considered and included as one reference.*
*Shie, C.-L., W.-K. Tao, and J. Simpson, 2006: A note on the relationship between temperature and water vapor over tropical oceans, including sea surface temperature effects, Special Issue of Advances in Atmospheric Sciences, Vol. 3, No. 1, 141-148. doi: 10.1007/s00376-006-0014-5*
*Electronic copy may be accessible at*
*http://www.iapjournals.ac.cn/fileDQKXJZ/journal/article/dqkxjz/2006/1/PDF/231scl.pdf*

Thank you for pointing us to this paper, we have included it into the manuscript!

---

## Author Comment (AC4) · 30 Oct 2020

In the course of the discussion on non-zero global total VIMD, Paul Berrisford and Michael Mayer (both ECMWF) recommended the use of the vertical integral of divergence of moisture flux (VIWVD) instead of, or in addition to, VIMD. VIWVD is similar to VIMD, but it is calculated from hourly instantaneous reanalysis fields, hence in spectral space. We verified that globally, VIWVD is a factor of $10$ closer to zero than VIMD, and the remaining deviation from zero is due to our use of $1°$ x $1°$ interpolated fields. Moreover, over-ocean VIWVD is very close to $(E - P)_{ocean}$, and the ocean and land totals are in agreement with other estimates in Table 4.

[Figure]

Nevertheless, we decided not to substitute VIMD by VIWVD in the manuscript, because VIMD is a better-known variable appearing in various studies. Moreover, VIMD is computed in the same way as E and P (i.e., from forecasts). Lastly, we think it is important to users of ERA5 VIMD to know its limitations and alternatives.

To acknowledge these facts, we changed the statement on lines 497–499 to:

"As observed above, the fact that ERA5 VIMD is calculated in grid point space causes global total $\nabla \cdot (vq)$ to be about $10 \cdot 10^3$ km$^3$ yr$^{-1}$, and not zero. In addition, due to the tighter observational control over land, analysis increments may be larger over ocean than over land and may cause net $\nabla \cdot (vq)$ to be close to net E-P over land, but less so over ocean (P. Berrisford, personal communication, Oct. 2020). There is another field in the ERA5 archive, the vertical integral of divergence of moisture flux (VIWVD, parameter ID p84.162), which is very similar to VIMD but is computed from hourly instantaneous reanalysis fields (and contains no contributions from liquid or solid water — but these can be neglected for our purposes). Globally VIWVD adds up to $0.9 \cdot 10^3$ km$^3$ yr$^{-1}$ ($0.003$ mm d$^{-1}$]), a factor of $10$ smaller than total VIMD. In addition, the agreement between over-ocean VIWVD and $(E - P)_{ocean}$ is much better than that found for VIMD and $(E - P)_{ocean}$, and at $41.6 \cdot 10^3$ km$^3$ yr$^{-1}$ and $-40.7 \cdot 10^3$ km$^3$ yr$^{-1}$, respectively, over-ocean VIWVD and over-land VIWVD are also in agreement with other values in the five right-most columns of Table 4."

And added the following recommendation to line 551 in the discussion section:

"Global total VIMD, however, does not equal zero, which is due to the numerical method used to compute VIMD. For studies of the global water cycle using ERA5 data, we recommend the use of VIWVD instead, as its global total is closer to zero and its totals over land and ocean are in better agreement with each other and with results from our and previous studies (Table 4)."

---

## Author Response (AR1)

Dear Editor,

Thank you for accepting our paper! The changes we made to the manuscript, most of which were inspired by the referees' helpful comments and suggestions, are described in some detail in our replies to the referees. Here, we only list the most significant changes.

First of all: we decided to update the outdated SEAFLUX version 2 data to version 3, which became publicly available shortly after the submission of our manuscript. We believe that performing the inter-comparison with the newest algorithm versions is of most use to the community. And although SEAFLUX3 evaporation data compares better with the other data sets, the satellite ensemble mean (SEM), to which we compared all data, was only marginally affected by the change in data sets. In particular, the general outcome and conclusions of our study remain the same.

Second: the deviation from zero that we found for global totals of ERA5 VIMD ignited a discussion between the authors and Michael Mayer, Paul Berrisford and Anton Beljaars from ECMWF. We took up their suggestion to include a statement about VIWVD, a parameter that is very similar to VIMD but is calculated in spectral space and which adds up to zero globally. Yet, we decided to primarily show results from VIMD, because this parameter is better known in the community and is calculated on the same grid as E and P.

More minor changes to the manuscript include: (i) adding a paragraph to the introduction to motivate our study from a climate change perspective and a statement to the discussion regarding the possible application of the presented framework to climate model intercomparisons. (ii) Updating all figures and tables following the switch to SEAFLUX version 3, and the slight change to the calculation of ERA5 $\Delta W$. We re-made Fig. 1 with more consistent color scales and improved the readability of Fig. 3 by moving the uncertainty information out of the plots. (iii) Changing the inaccurate notation of $\nabla Q$ to $\nabla \cdot (vq)$ throughout the manuscript and (iv) adding the literature references suggested by the referees, in particular adding the numbers from Allan et al. (2020) to our Table 4.

The changes have improved the manuscript and we gratefully acknowledge the editor and the referees for their suggestions and comments.

Please find the revised manuscript with marked-up changes below.

Kind regards,

Marloes Gutenstein and co-authors

[revised manuscript text omitted]